# Source-Free and Image-Only Unsupervised Domain Adaptation for Category Level Object Pose Estimation

**Prakhar Kaushik  Aayush Mishra  Adam Kortylewski[†]  Alan Yuille**
Johns Hopkins University
[†]University of Freiburg and Max-Planck-Institute for Informatics
{pkaushi1,amishr24,ayuille1}@jh.edu  [†]akortyle@mpi-inf.mpg.de

## Abstract

We consider the problem of source-free unsupervised category-level pose estimation with only images to a target domain without any access to source domain data or 3D annotations during adaptation. Collecting and annotating real-world 3D data and corresponding images is laborious, expensive, yet unavoidable process, since even 3D pose domain adaptation methods require 3D data in the target domain. We introduce 3DUDA, a method capable of adapting to a nuisance ridden target domain without 3D or depth data. Our key insight stems from the observation that specific object subparts remain stable across out-of-domain (OOD) scenarios, enabling strategic utilization of these invariant subcomponents for effective model updates. We represent object categories as simple cuboid meshes, and harness a generative model of neural feature activations modeled at each mesh vertex learnt using differential rendering. We focus on individual locally robust mesh vertex features and iteratively update them based on their proximity to corresponding features in the target domain even when the global pose is not correct. Our model is then trained in an EM fashion, alternating between updating the vertex features and the feature extractor. We show that our method simulates fine-tuning on a global pseudo-labeled dataset under mild assumptions, which converges to the target domain asymptotically. Through extensive empirical validation, including a complex extreme UDA setup which combines real nuisances, synthetic noise, and occlusion, we demonstrate the potency of our simple approach in addressing the domain shift challenge and significantly improving pose estimation accuracy.

## 1 Introduction

In recent years, object pose estimation has witnessed remarkable progress, revolutionizing applications ranging from robotics (Du et al., 2019; Wang et al., 2019a; Wong et al., 2017; Zeng et al., 2017) and augmented reality (Marchand et al., 2016; Marder-Eppstein, 2016; Runz et al., 2018) to human-computer interaction. There have been works for 3D and 6D pose estimation that have focused primarily on instance-level (He et al., 2021; 2020; Park et al., 2019; Peng et al., 2019; Tremblay et al., 2018; Wang et al., 2019a; Xiang et al., 2018) pose estimation methods. However, these methods require object-specific 3D CAD models or instance-specific depth information and are often unable to estimate object pose without given instance-specific 3D priors. Category-level methods (Chen et al., 2020a; Chen & Dou, 2021; Lin et al., 2021; Tian et al., 2020; Wang et al., 2019b; 2021b) are more efficient, but often still require some 3D information, such as the ground truth depth map (Wang et al., 2019b; Lin et al., 2021; Lee et al., 2022) or point clouds (Lee et al., 2023). Acquiring such labeled 3D data across different domains is often a formidable challenge, impeding the performance of these models when deployed in real-world scenarios. Few recent attempts (Lee et al., 2022; 2023) to perform category level pose estimation in a semi-supervised manner also require a ground truth depth map (Lee et al., 2023) or point cloud (Lee et al., 2022) for every instance.

In this paper, we focus on ameliorating the aforementioned drawbacks in UDA for 3D pose estimation. We design a model that is capable of adapting to a target domain in an unsupervised manner **without requiring any kind of 3D data and using only RGB images in the target domain**. Our

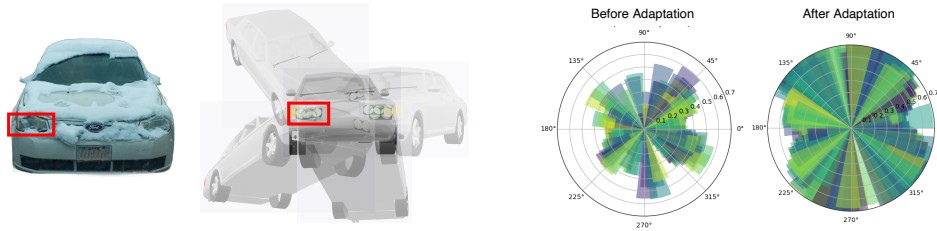

(a) Local Pose Ambiguity          (b) Local Part Robustness

Figure 1: Our method utilizes two key observations- (a) **Local Pose Ambiguity**, i.e. the inherent pose ambiguity that occurs when we can only see a part of the object. We utilize this ambiguity to update the local vertex features which roughly correspond to object parts, even when the global pose of the object may be incorrectly estimated. (b) **Local Part Robustness** refers to the fact that certain parts (e.g. headlights in a car) are less affected in OOD data, which is verified by the (azimuth) polar histogram representing the percentage of robustly detected vertex features per image in target domain (OOD-CV (Zhao et al., 2023)) using the source model (*Before Adaptation*). Even before adaptation, there are a few vertices which can be detected robustly and therefore are leveraged by our method to adapt to the target domain as seen by the increased robust vertex ratio *After Adaptation*.

model does not require source data during adaptation and as shown later, is capable to adapt to real world OOD nuisances (Zhao et al., 2023) without requiring any synthetic data or augmentations.

Our source model is based on the idea of generative modeling of neural network features (Kortylewski et al., 2020; Wang et al., 2021a; 2023; Ma et al., 2022) that have been used to perform category-level 3D and 6D pose estimation. However, all of these methods are fully supervised and cannot be trivially adapted to an OOD target. We extend these neural feature-level render-and-compare methods' capabilities for source-free unsupervised learning, which can be utilized in real-world OOD scenarios. Our method, 3DUDA, is based on the observation that **certain object subparts remain stable and invariant across out-of-domain (OOD) scenarios**, as seen in Figure 1, thereby offering a robust foundation for model updates. We utilize this ensemble of less modified object local subparts and their inherent pose ambiguity in the nuisance-ridden target domain images to adapt the source model. This allows us to ignore noise-ridden global object pose and still obtain relevant information from more robust local sub-components of the object. We focus on individual mesh vertex features, iteratively updating them based on their proximity to the corresponding features in the target domain. Our experiments show that this simple idea allows us to perform robust pose estimation in OOD scenarios with only images from the target domain.

In summary, we make several important contributions in this paper.

1. We introduce 3DUDA - which is (in our knowledge) the first method to do **image only, source free unsupervised domain adaptation for category level 3D pose estimation**.

2. 3DUDA utilizes local pose ambiguity and their relative robustness to adapt to nuisance ridden domains without access to any 3D or synthetic data. We present theoretical analysis for this insight, which motivates our method.

3. We evaluate our model on real world nuisances like shape, texture, occlusion, etc. as well as image corruptions and show that our model is able to adapt robustly in such scenarios. Our method performs exceedingly well in *extreme UDA* setups where multiple nuisance factors such as real-world nuisance, synthetic noise, and partial occlusion are combined.

## 2 RELATED WORKS

**Category-level 3D pose estimation** is a task for estimating the 3D pose of unseen instances but in a known category. Current pose estimation approaches can be divided into keypoint approaches, which utilize semantic keypoints on 3D objects to predict 3D pose (Pavlakos et al., 2017a; Zhou et al., 2018) and render and compare methods which predict pose by fitting a 3D rigid transformation at the image level (Chen et al., 2020b; Wang et al., 2019b) or feature level (Wang et al., 2021a).

**Feature-level render-and-compare** methods predict 3D pose by minimizing reconstruction error between predicted and rendered (e.g. from a 3D mesh and a corresponding pose) object representations. Such optimization often helps in avoiding complex loss landscapes that arise by doing

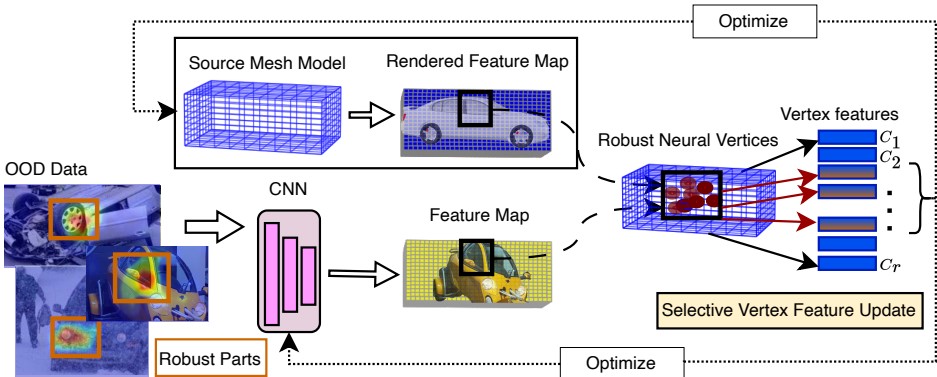

Figure 2: Overview of Our Method (3DUDA)

(a) We extract neural features from source model CNN backbone $f_i = \phi_w(\mathcal{X}_\mathcal{T})$ and render feature maps from the source mesh model ($\mathfrak{M}_\mathcal{S}$) (using vertex features $C_r$) and the pose estimate is optimized using render-and-compare (b) For this incorrectly estimated global pose, we measure similarity of every individual visible vertex feature with the corresponding image feature vector in $f_i$ *independently* (Equation 3) and update individual vertex features using average feature vector values for a batch of images (Equation 4). (c) The mesh model is then updated using these changed vertices and the backbone is optimized using the optimized neural mesh.

render-and-compare at the image pixel level. (Wang et al., 2019b) predict the pose of the object by solving a rigid transformation between the 3D model $\mathfrak{M}$ and the NOCS maps with the Umeyama algorithm (Pavlakos et al., 2017b). (Iwase et al., 2021) learned features using differentiable Levenberg-Marquardt optimization, whereas (Wang et al., 2021a; Ma et al., 2022) learned contrastive features for the 3D model $\mathfrak{M}$ and utilized a similar render-and-compare setup.

**Unsupervised Domain Adaptation for 3D pose estimation** Unfortunately, all the methods mentioned above are fully supervised. However, there are some semi-supervised methods like (Fu & Wang, 2022; Peng et al., 2022) that often require labeled target domain image and 3D data to work. Even methods like (Lee et al., 2022; 2023) require instance depth data, point cloud and segmentation labels during test-time inference. Other methods like (Yang et al., 2023) create synthetic data and mix them with some amount of annotated real data in order to do Synthetic to Real semi-supervised domain adaptation. To the best of our knowledge, there is no previous work on unsupervised 3D pose estimation which is source-free and requires only images for adaptation. Additionally, there is also a dearth of work in 3D pose estimation which performs UDA for real-world nuisances like changes in texture, weather, etc. and in the presence of problems like occlusion.

## 3 METHODOLOGY

An overview of our unsupervised domain adaptation method, 3DUDA can be found in Figure 2. After defining the notation and setup, we review our feature-level neural render-and-compare source model in Section 3.1 before describing our method in detail in Section 3.2.

**Notation** For each object category $y$, we define three sets of parameters: a CNN backbone $\Phi_w$, a neural mesh $\mathfrak{M}$, and a clutter model $\mathcal{B}$. We denote the neural feature representation of an input image $\mathcal{X}$ as $\Phi_w(\mathcal{X}) = F^a \in \mathcal{R}^{H \times W \times d}$. Where $a$ is the output of layer $a$ of a deep convolutional neural network backbone $\Phi_w$, with $d$ being the number of channels in layer $a$. $f_i^a \in \mathcal{R}^d$ is a feature vector in $F^a$ at position $i$ on the 2D lattice $P$ of the feature map. We drop the superscript $a$ in subsequent sections for notational simplicity. We represent our supervised source domain model with subscript $\mathcal{S}$ and our unsupervised target domain model with subscript $\mathcal{T}$. For more details, see A.1.

### 3.1 SOURCE MODEL: POSE-DEPENDENT FEATURE LEVEL RENDER AND COMPARE

Our source model is similar to previous work like Wang et al. (2021a; 2023); Ma et al. (2022) on category-level pose estimation using neural feature level render and compare. These methods themselves are 3D extensions of feature generative models such as Kortylewski et al. (2020). Our source model defines a probabilistic generative model of normalized real-valued feature activations

$F$ conditioned on a 3D neural mesh representation $\mathfrak{M}$. The neural mesh model aims to capture the 3D information of the foreground objects. For each object category $y$, the source model defines a neural mesh $\mathfrak{M}_\mathcal{S}$ as $\{\mathcal{V}, \mathcal{C}\}$, where $\mathcal{V} = \{V_r \in \mathbb{R}^3\}_{r=1}^R$ is the set of vertices of the mesh and $\mathcal{C} = \{C_r \in \mathbb{R}^c\}_{r=1}^R$ is the set of learnable features, i.e. neural features. $r$ denotes the index of the vertices. $R$ is the total number of vertices. We also define a clutter model $\mathcal{B} = \{\beta_n\}_{n=1}^N$ to describe the backgrounds. $N$ is a prefixed hyperparameter. For a given object pose or camera viewpoint $g$, we can render the neural mesh model $\mathfrak{M}_\mathcal{S}$ into a feature map using (differentiable) rasterization (Kato et al., 2020). We can compute the object likelihood of a target feature map $F \in \mathcal{R}^{H \times W \times D}$ as

$$p(F|\mathfrak{M}, g, \mathcal{B}) = \prod_{i \in \mathcal{FG}} p(f_i|\mathfrak{M}, g) \prod_{i' \in \mathcal{BG}} p(f_{i'}|B), \qquad (1)$$

where $\mathcal{FG}$ and $\mathcal{BG}$ denote the foreground and background pixels, respectively. $\mathcal{FG}$ is set of all the positions in the 2D lattice $P$ covered by the mesh $\mathfrak{M}$ and $\mathcal{BG}$ are the positions that are not. We define $P(f_i|\mathfrak{M}(V_r, C_r), g) = Z[\kappa_r] \exp\{\kappa_{r,c} f_i \cdot n_{r,c}\}$ as a von Mises Fisher (vMF) distribution with mean $C_r$ and concentration parameter $\kappa_r$. For more details, please refer to the Appendix A.7.1.

### 3.2 3DUDA: Unsupervised Learning of 3D pose using neural feature synthesis and selective vertex feature update

Given only the source model $\mathcal{S}$ (and no source data $\mathcal{X}_\mathcal{S}$) and some non-annotated target domain RGB images, we adapt $S$ to be able to perform well on the target domain. Figure 3 (NeMo column) shows examples of the performance of the source model in an OOD scenario. As expected, the estimated pose has diverged significantly from the ground truth pose in the target domain, indicating that the feature generative model parameterized by the neural mesh model $\mathfrak{M}_\mathcal{S}$ is no longer an adequate representative of the same object *as a whole*.

However, a crucial observation, as seen in Figure 1, is that although the neural mesh model may not be a good *global model* for an object $y$, there is still a subset of robust vertices that corresponds to parts of objects that have undergone less changes in the new domain. Intuitively, this can be understood as some parts of an object changing less across domains. The number of such vertices and the threshold within which their shift is contained are functions of the domain nuisance variables and the object itself. This property is intuitively leveraged by humans regularly to adapt their previous knowledge to understand new, unseen objects. For example, a car that may undergo changes involving shape, context, and texture will still have parts such as wheels, headlights, and windshield that change less or none at all across these domain shifts. We leverage this observation to adapt the source model in an unsupervised manner.

#### Neural feature synthesis with Multi Pose Initialization

A fundamental benefit of Neural Mesh Models like our source model is that they are generative at the level of neural feature activations. This makes the overall reconstruction loss very smooth compared to related works that are generative on the pixel level (Wang et al., 2021a; 2023). Therefore, the source model can be optimized w.r.t. the pose parameters with standard stochastic gradient descent and contains one clear loss global optimum. This is no longer true in an OOD scenario.

**Neural Feature Rendering** For inference with the source model, we can infer the 3D pose $g$ of the object $y$ by minimizing the negative log likelihood of the model. Specifically, we first extract the neural features of the image $F = \Phi_w(\mathcal{X})$ from the source CNN backbone. We define an initial pose $g_{init}$ using random initialization or by pre-rendering and comparing some pose samples. Using the initial pose, we render the neural mesh $\mathfrak{M}$ into a feature map $F' \in \mathcal{R}^{H \times W \times D}$. The projected feature map is divided into $\mathcal{FG}$ and $\mathcal{BG}$, depending on which pixels in the feature map are covered by the projected mesh features. We compare the rendered feature map and the image feature map position-wise. Given that the feature vectors are normalized and considering a constant $\kappa$, the loss can be refactored as a simple reconstruction loss. The dot products are normalized accordingly.

$$\mathcal{L}_{rec} = 1 - \ln p(F|\mathfrak{M}, g, \mathcal{B}) = 1 - \left(\sum_{i \in \mathcal{FG}} f_i * f_i' + \sum_{j \in \mathcal{BG}} f_j * \beta\right) \qquad (2)$$

The pose $g_{init}$ is optimized by minimizing Equation 2 using stochastic gradient descent.

**Multi-Pose Initialization** Figure 6 shows different render-and-compare optimized pose estimates by the source model that are optimized from different initial 3D pose in the target domain. This happens because optimization is stuck in different local optima in the OOD loss landscape. Therefore, instead of random pose initialization, we pre-render a uniform sampling of poses from the neural mesh model and compare them with the image features. $1-5$ initial poses are chosen dependent how similar they are to the feature map and how far away they are from each other. We then optimize these initial poses using Equation 2, and may end up with multiple final rendered feature maps and estimated poses as shown in Figure 6. Even though the estimated object poses may be incorrect, we can still utilize these rendered maps for our Selective Vertex Feature Adaptation.

### 3.2.1 PROGRESSIVE SELECTIVE LOCAL VERTEX FEATURE ADAPTATION

**Local Vertex-Feature Similarity** We define the similarity between an individual rendered neural mesh $\mathfrak{M}$ vertex feature $C_r$ and its corresponding CNN feature $f$ (shown as $f_{i \to r}$) for a pose $g$ given a renderer $\mathfrak{R}$ as a function of the parametric vMF score/likelihood (Du et al., 2022);

$$\mathcal{L}_{sim}(f_{i \to r}, C_r) = Z[\kappa_r] \exp\left(\kappa_r f_{i \to r}^T C_r\right), \qquad \forall i \in \mathcal{FG}, \quad C_r = \mathfrak{R}(\mathfrak{M}, g) \tag{3}$$

Similar similarity measures have been shown to be robust OOD detectors in earlier works like (Du et al., 2022). We define a rejection criteria dependent on a threshold $\delta$ such that all individual feature vectors from multiple data samples that correspond to a specific vertex feature $C_r$ considered OOD if $\mathcal{L}_{sim}(f_{i \to r}, C_r) < \delta_r$ and are not used to update these features. We can choose the threshold $\delta_r$ st. majority (90-95%) of source domain features lie within the likelihood score.

**Selective Vertex Adaptation (SVA) from Rendered Neural Features** For a batch (size n) of target domain images $\mathcal{X}_{\mathcal{T},i}$, we obtain their neural features from the finetuned CNN backbone $\phi_w$ and we render neural feature maps $F_i$ from our mesh model $\mathfrak{M}_S$ using differentiable render and compare from initial pose estimates $g_{init}$. We then spatially match the similarity of every rendered vertex feature with its corresponding image feature *independently*. For every vertex feature $C_r$, we average the corresponding image features $f_i^a$ at position $a$ in the 2D lattice which are above a threshold hyperparameter $\delta$ using Equation 3, we update a neural vertex feature $C_r$ as follows:

$$C_r^{t+1} \leftarrow \alpha C_r^t + (1-\alpha)\frac{1}{n}\sum_n f_{i \to r}^a, \qquad \forall f_i^a \ni \mathcal{L}_{sim}(C_r, f_{i \to r}^a) > \delta_r \tag{4}$$

where $C_r^t$ is the current vertex feature at timestep $t$ and $C_r^{t+1}$ is the updated vertex feature. $\alpha$ is a moving average hyperparameter. This can be done for the entire target domain adaptation data or in a batched manner. Subsequently, the estimated pose $g'$ is recalculated with the updated neural mesh model, and the CNN backbone is updated by gradient descent iteratively with the following loss;

$$\mathcal{L} = \sum_{r \in R_v} \log \frac{Z[\kappa_r]e^{\kappa f_{i \to r}C_r}}{\sum_{l \in R, l \notin \mathcal{N}_r} Z[\kappa_l]e^{\kappa f_{i \to l}C_l} + \sum_{n=1}^{N} Z[\kappa_n']e^{\kappa' f_{i \to n}\beta_n}}, \tag{5}$$

where $R_v$ denotes all visible vertices for the input image $\mathcal{X}$. $\mathcal{N}_r$ denotes the vertices near $r$. We iteratively update subsets of vertex features and finetune the CNN backbone till convergence in a EM type manner. In practice, to avoid false positives and encourage better convergence, we establish a few conditions on our selective vertex feature adaptation process. To save computational overhead, we can fix $\kappa$ for the loss calculation . We fix a hyperparameter $\psi_n$ that controls the least number of local vertices detected to be similar ($5-10\%$ of visible vertices). We also drop samples with low global similarity values during the backbone update. $\kappa_r$ can also be recalculated in each time step $t$ using the updated $C_r^t$ for a more robust measurement of similarity.

**Unsupervised 3D pose estimation using SVA** Figure 2 gives an intuition for our method, 3DUDA. It works as follows: (1) Extract neural features from source model CNN backbone $f_i = \phi_w(\mathcal{X}_\mathcal{T})$ (2) Pre-render feature maps from the source mesh ($\mathfrak{M}_S$) (using vertex features $C_r$) and compare with image features (3) Choose top-3 similar rendered feature maps and calculate the optimized pose using gradient descent with reconstruction loss w.r.t $f_i$ (Equation 2). (4) For an optimized pose, measure the similarity of every individual visible vertex feature with the corresponding vector in $f_i$ *independently* (Equation 3). (5) Update individual vertex features using average feature vector values for a batch of images (Equation 4). (6) Finetune CNN backbone using rendered feature maps obtained from Steps 1-3 using the updated vertex features (7) Continue steps 5 and 6 iteratively until convergence. (8) Post adaptation, evaluate 3D object pose of target test data using steps 1-3.

## 3.3 THEORETICAL RESULTS

Prior UDA works attempt to adapt in the target domain using a global pseudo-labeling setup. In global pseudo-labeling, a subset of images from the target domain are identified for which a majority (typically $> R\Omega$ for some $\Omega \in [0,1]$. We use $\Omega = 1$ for analysis.) of the visible vertex features satisfy $\mathcal{L}_{\text{sim}}(f_{i\to r}, C_r) > \delta_r$. These images are used to fine-tune the model. However, depending on how different the target domain is from the source domain, a very small fraction of the target domain images (even none sometimes) usually satisfies this property in real datasets. This leads to negligible domain adaptation. However, upon careful inspection, we observed a robust subset of vertices in the neural meshes produced by the source-trained model. And as the features corresponding to these robust vertices are assumed independent, they can be used to fine-tune the model. In this section, we present conditions under which SVA simulates fine-tuning on a global pseudo-labeled dataset. This also reveals that the standard global pseudo-labeling setup is a special case of this formulation.

Let the distribution of a rendered vertex feature $C_r$ be denoted by $P^r$. Let the joint distributions of these rendered vertex features in the source and target domain be denoted by $P_S = \prod_r P_S^r$ and $P_T = \prod_r P_T^r$ respectively. They are written as products of the individual marginal distributions because of the standard independence assumption between the vertices. Note that the distribution $P_S$ is an approximation of the true underlying distribution $P_S^*$. This approximate distribution is achieved by training the source model $\mathcal{S}$ on a finite i.i.d. sample $\mathcal{X}_S$ from $P_S^*$ and their corresponding labels (ground-truth poses) $\mathcal{Y}_S$. Similarly, the i.i.d. sample $\mathcal{X}_T$ elicits an approximate distribution $P_T$ of the true $P_T^*$. Adapting for $P_T$ is challenging because we do not have the corresponding $\mathcal{Y}_T$.

**Definition 3.1** *(Vertex $K$-partition) A vertex $K$-partition is defined as a partition of the set of vertices (indexed by $r \in \{1, 2, ..., R\}$) into $K$ non-empty mutually disjoint subsets (indexed by $k \in \{1, 2, ..., K\}$). Let the set of vertices in each partitioned subset be denoted by $I_k$.*

A given vertex $K$-partition would split the joint distribution $P_S$ into $K$ independent joint distributions (denoted by $P_S^{I_k} = \prod_{r \in I_k} P_S^r$) such that $P_S = \prod_k P_S^{I_k}$. The same extends for the corresponding target domain distributions.

**Definition 3.2** *($k\delta$-subset) For a given sample $\mathcal{X}$ and vertex $K$-partition, a $k\delta$-subset is defined as $\mathcal{X}^{k\delta} \subseteq \mathcal{X}$ such that, $\mathcal{L}_{sim}(f_{i\to r}, C_r) > \delta_r \, \forall \, r \in I_k$. The corresponding approximation of $P^{I_k}$ under $\mathcal{X}^{k\delta}$ is denoted by $P^{I_k\delta}$.*

**Assumption 3.3** *(Piece-wise Support Overlap) There exists a vertex $K$-partition such that the $k\delta$-subset of the target sample $\mathcal{X}_T$ satisfies,*

$$|\mathcal{X}_T^{k\delta}| \neq 0 \,\forall k \in \{1, 2, ..., K\}$$
$$\text{and as } |\mathcal{X}_T^{k\delta}| \to \infty, \prod_k P_T^{I_k\delta} \to P_T^*$$

This assumption requires the joint distribution of partitioned vertex subsets under $\mathcal{X}^{k\delta}$ to asymptotically approximate the corresponding true distributions. Intuitively, this translates to having enough support in the target domain such that samples satisfying the similarity constraint (Equation 3) in each $k\delta$-subsets approximates the true target distribution of that vertex partition set.

**Theorem 3.4** *A target domain $\mathcal{X}_T$ satisfying assumption 3.3, elicits another target domain $\mathcal{X}_T^e$ such that each sample in $\mathcal{X}_T^e$ satisfies the global-pseudo labelling constraint ($\mathcal{L}_{sim}(f_{i\to r}, C_r) > \delta_r \,\forall\, r \in \{1, 2, ..., R\}$). Asymptotically with the size of the domain, $\mathcal{X}_T^e \to \mathcal{X}_T$.*

The proof of this theorem is by construction of the set $\mathcal{X}_T^e$ for any $\mathcal{X}_T$, and has been defered to the appendix A.3. It is easy to see that a global-pseudo labellelling setup is the special case of this formulation when the vertex $K$-partition in assumption 3.3 is the trivial partition with $K = 1$. It is also noteworthy that in the asymptotic case, even the global labelling setup would yield the same adaptability as SVA; but in practice, SVA yields much more data for adaptation than the former (see figure 5). Although the elicited target domain $\mathcal{X}_T^e$ does not represent the true target distribution precisely, it does yield more adaptability than global-pseudo labelling for a finite-sample and under assumption 3.3, asymptotically adapts to the true distribution (Figure 4).

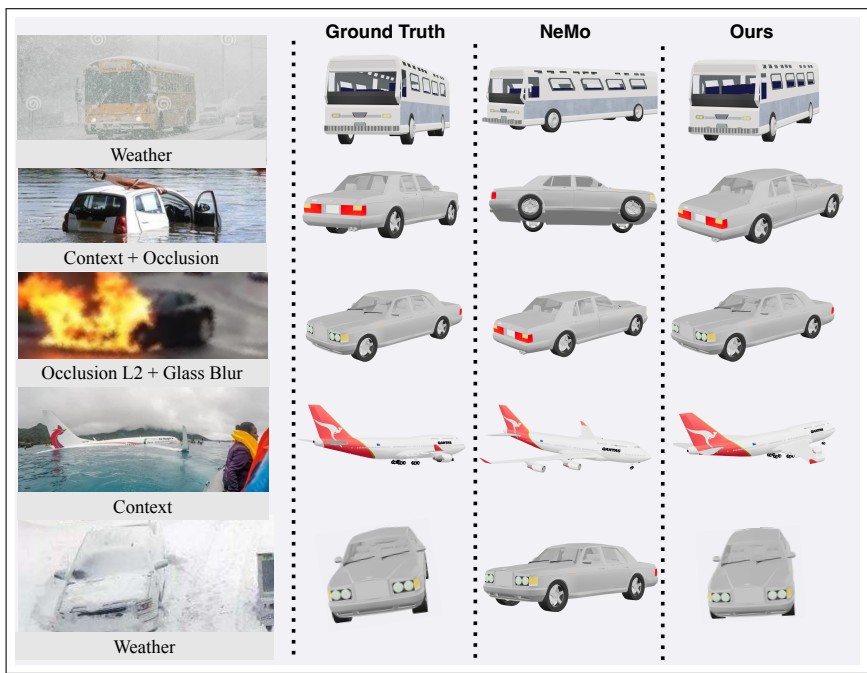

Figure 3: Qualitative Results of 3DUDA compared to ground truth and NeMo (Wang et al., 2021a). 3DUDA adapts to real world OOD target domains consisting of nuisances like weather and occlusion in an unsupervised manner and produces robust 3D object pose estimates. The CAD objects are for representation only and are taken from ShapeNet (Chang et al., 2015).

## 4 EXPERIMENTS

**Data** We evaluate our model on OOD-CV (Zhao et al., 2023) and Imagenet-C (Hendrycks & Dietterich, 2019) corrupted Pascal3D+ dataset (Xiang et al., 2014). OOD-CV is a benchmark introduced to evaluate the robustness of the model in OOD scenarios. It includes OOD examples of 10 categories that cover unseen variations of nuisances including pose, shape, texture, context, and weather. The source model is trained on IID samples while the model is adapted and evaluated on OOD data for individual and combined nuisances. For Corrupted-Pascal3d+, we corrupt the adaptation and evaluation data with synthetic corruptions like shot noise, elastic deformation, fog, etc. from Imagenet-C. PASCAL3D+ dataset contains objects from 12 man-made categories, and each object is annotated with 3D pose, 2D centroid, and object distance. During adaptation and inference, only RGB images are provided. For a harder set-up of real-world nuisances combined with partial occlusion, we test our algorithm on Occluded-OOD-CV dataset which has been created in a manner similar to Wang et al. (2020) with 2 levels of object occlusion (L1(20−40%), L2(40−60%)).

We also evaluate our model on two **extreme** UDA setups of (1) Real + Synthetic corruptions. We add Imagenet-C corruptions to the OOD-CV dataset and expect models to adapt from clean IID source data to these data in an unsupervised manner. (2) Real + Synthetic Corruption + Partial Occlusion. These are very difficult scenarios for UDA which, to our knowledge, have not been attempted before in a semi-supervised or unsupervised 3D pose estimation.

**Metrics** 3D pose estimation aims to recover the 3D rotation parameterized by azimuth, elevation, and in-plane rotation of the viewing camera. We follow previous works like Zhou et al. (2018); Wang et al. (2021a); Ma et al. (2022) and evaluate the error between the predicted rotation matrix and the ground truth rotation matrix: $\Delta(R_{pred}, R_{gt}) = \frac{||logm(R_{pred}^T R_{gt})||_{\mathcal{F}}}{\sqrt{2}}$. We report the accuracy of the pose estimation under common thresholds, $\frac{\pi}{6}$ and $\frac{\pi}{18}$ alongwith median error.

**Implementation Details** An Imagenet pretrained Resnet50 is used as a feature extractor for our source model. The cuboid mesh is defined for each category and ensures that the majority of object area are covered by it. The source model is trained for 800 epochs with a batch size of 32 using

an Adam optimizer in a fully supervised manner. For every adaptation step, we require a minimum batch size of 32 images for selective vertex and feature extractor update. We can also set the batch size for a step adaptively by requiring enough samples s.t. $\simeq 80\%$ of vertices can be updated. For a fixed $\kappa$, our local vertex feature similarity threshold is .8. We train our model by switching between (in an EM fashion) selective vertex update and feature extractor training for about 100 epochs. Our adaptation model is implemented in PyTorch (with PyTorch3D for differential rasterization) and takes around 3 hours to train on 2 A5000 GPUs.

**Baseline Models** We evaluate NeMo (Wang et al., 2021a), DMNT (Wang et al., 2023), SyntheticP3D (Yang et al., 2023) and standard Resnet-50. Wang et al. (2021a; 2023); Yang et al. (2023) are pose estimation methods which utilize similar feature level render and compare methodology as our source model and have been shown to be robust and efficient. (Res50-General) is a ResNet50 classifier that formulates the pose estimation task for all categories as one single classification task. Note that all models are evaluated on image only unsupervised learning setup. Although annotated target domain images are not provided to models, data augmentation or synthetic data is allowed for our baseline models.

## 4.1 UNSUPERVISED 3D POSE ESTIMATION: RESULTS AND ANALYSIS

**OOD-CV** Table 1 shows the Unsupervised 3D pose estimation results on OOD-CV (Zhao et al., 2023) dataset. This is our primary result, which shows our model efficacy in real-world scenarios. A qualitative comparison with Wang et al. (2021a) can be seen in Figure 3. All compared methods suffer equally in the real OOD scenarios. Surprisingly, the general Resnet50 model performs quite well relative to more complex models like NeMo and DMNT, suggesting that the additional category data is helpful in OOD scenarios. However, our method clearly outperforms all the models and is able to significantly bridge the domain gap.

Table 1: Unsupervised 3D Pose Estimation for OOD-CV (Zhao et al., 2023) dataset

| Nuisance | Combined | shape | $\frac{\pi}{6}$**Accuracy**⬆ | | | |
|---|---|---|---|---|---|---|
| | | | pose | texture | context | weather |
| Res50-General | 51.8 | 50.5 | 34.5 | 61.6 | 57.8 | 60.0 |
| NeMo (Wang et al., 2021a) | 48.1 | 49.6 | 35.5 | 57.5 | 50.3 | 52.3 |
| MaskRCNN (He et al., 2018) | 39.4 | 40.3 | 18.6 | 53.3 | 43.6 | 47.7 |
| DMNT (Wang et al., 2023) | 50.0 | 51.5 | 38.0 | 56.8 | 52.4 | 54.5 |
| P3D (Yang et al., 2023) | 48.2 | 52.3 | 45.8 | 51.0 | 54.6 | 44.5 |
| **Ours** | **94.0** | **93.7** | **95.1** | **97.0** | **95.5** | **83.1** |
| | | | $\frac{\pi}{18}$**Accuracy**⬆ | | | |
| Res50-General | 18.1 | 15.7 | 12.6 | 22.3 | 15.5 | 23.4 |
| NeMo (Wang et al., 2021a) | 21.7 | 19.3 | 7.1 | 33.6 | 21.5 | 30.3 |
| MaskRCNN (He et al., 2018) | 15.3 | 15.6 | 1.6 | 24.3 | 13.8 | 22.9 |
| DMNT (Wang et al., 2023) | 23.6 | 20.7 | 12.6 | 32.6 | 16.6 | 33.5 |
| P3D (Yang et al., 2023) | 14.8 | 16.1 | 12.3 | 16.6 | 12.1 | 16.3 |
| **Ours** | **87.8** | **82.1** | **69.5** | **92.6** | **89.3** | **90.7** |

**Pascal3D→Corrrupted-Pascal3D+** Table 2 shows the Unsupervised 3D pose estimation results on this setup for multiple corruptions. As expected, the drop in model performance is largely dependent on the type of corruption and its severity. Our method still performs significantly better when dealing with synthetic corruptions.

**Occluded-OOD-CV** Table 3 (OccL1/L2) shows the Unsupervised 3D pose estimation results on Occluded-OOD-CV dataset at 2 levels of partial occlusion. This is a harder setup in which real nuisances are combined with occlusion. Our method is able to perform exceedingly well even in such a complex target domain with upto $67\%$ improvement in accuracy. This is because our selective vertex adaptation focuses independently on adapting individual neural vertices (and the model) and is able to ignore occluded vertices for adaptation. Notably, Wang et al. (2021a) has been shown to be robust to occlusion but suffers when it is combined with real world nuisances.

Table 2: Unsupervised 3D pose estimation results for Pascal3d+ $\rightarrow$ Corrupted-Pascal3D+ (Metrics : $\pi\backslash6$ Accuracy ($\frac{\pi}{6}$), $\pi\backslash18$ Accuracy ($\frac{\pi}{18}$), Median Error (Er))

| | $\frac{\pi}{6}$⬆ | $\frac{\pi}{18}$⬆ | Er⬇ | $\frac{\pi}{6}$⬆ | $\frac{\pi}{18}$⬆ | Er⬇ | $\frac{\pi}{6}$⬆ | $\frac{\pi}{18}$⬆ | Er⬇ | $\frac{\pi}{6}$⬆ | $\frac{\pi}{18}$⬆ | Er⬇ |
|---|---|---|---|---|---|---|---|---|---|---|---|---|
| | **Gaussian Noise** | | | **Shot Noise** | | | **Impulse Noise** | | | **Defocus Blur** | | |
| NeMo | 43.7 | 21.3 | 42.1 | 50.6 | 25.3 | 35.0 | 45.4 | 22.2 | 39.4 | 72.9 | 41.8 | 16.0 |
| Ours | **84.3** | **59.1** | **9.8** | **85.9** | **62.0** | **9.0** | **84.0** | **58.5** | **10.1** | **87.8** | **64.6** | **8.0** |
| | **Glass Blur** | | | **Motion Blur** | | | **Zoom Blur** | | | **Snow** | | |
| NeMo | 56.7 | 27.0 | 33.8 | 69.7 | 39.2 | 18.7 | 69.0 | 39.7 | 19.1 | 69.9 | 40.1 | 18.9 |
| Ours | **86.7** | **62.4** | **8.6** | **88.0** | **63.8** | **8.3** | **87.9** | **65.1** | **8.1** | **87.7** | **64.0** | **8.2** |
| | **Frost** | | | **Fog** | | | **Contrast** | | | **Elastic Transform** | | |
| NeMo | 73.3 | 44.1 | 16.4 | 85.5 | 59.0 | 9.5 | 74.5 | 43.8 | 14.7 | 77.4 | 50.3 | 13.8 |
| Ours | **86.3** | **62.5** | **8.6** | **88.7** | **65.6** | **7.8** | **88.8** | **66.7** | **7.6** | **88.2** | **64.4** | **8.1** |
| | **Pixelate** | | | **Speckle Noise** | | | **Gaussian Blur** | | | **Spatter** | | |
| NeMo | 77.5 | 53.0 | 13.0 | 67.9 | 38.3 | 20.8 | 68.3 | 36.5 | 18.7 | 72.4 | 44.2 | 17.2 |
| Ours | **88.7** | **65.4** | **7.8** | **87.7** | **64.2** | **8.0** | **87.7** | **63.8** | **8.3** | **87.5** | **63.9** | **8.4** |

NeMo (Wang et al., 2021a).

**Extreme UDA: Real+Synthetic Corruption**   Table 3 (OOD+SN/GB) show the results on the extreme UDA setup combining real and synthetic corruptions from the OOD-CV (Combined) and Imagenet-C datasets. We again see significant improvements compared to Wang et al. (2021a) which have been shown previously to be robust under this challenging setup.

Table 3: Unsupervised 3D pose estimation results for Occlusion and Extreme UDA setup

(a) **OccL1/L2**: Real Nuisance (OOD-CV (Combined)) + Occlusion (Level1/Level2) (b) **OOD+SN/GB**: Real Nuisance (OOD-CV) + Synthetic Noise (Speckle Noise/Glass Blur) (c) **L1/L2+Spec**: Real Nuisance (OOD-CV) + Occlusion (L1/L2) + Synthetic Noise (Speckle Noise)

| | OccL1 | | OccL2 | | OOD+SN | | OOD+GB | | L1+Spec | | L2+Spec | |
|---|---|---|---|---|---|---|---|---|---|---|---|---|
| | $\frac{\pi}{6}$ | $\frac{\pi}{18}$ | $\frac{\pi}{6}$ | $\frac{\pi}{18}$ | $\frac{\pi}{6}$ | $\frac{\pi}{18}$ | $\frac{\pi}{6}$ | $\frac{\pi}{18}$ | $\frac{\pi}{6}$ | $\frac{\pi}{18}$ | $\frac{\pi}{6}$ | $\frac{\pi}{18}$ |
| NeMo | 30.6 | 10.2 | 24.1 | 6.6 | 32.7 | 10.2 | 29.6 | 9.5 | 18.6 | 3.4 | 15.1 | 2.7 |
| Ours | **84.6** | **77.1** | **78.7** | **70.4** | **80.5** | **63.0** | **77.7** | **65.9** | **69.4** | **50.4** | **60.6** | **38.9** |

**Extreme UDA: Real+Synthetic Corruption+Occlusion**   Table 3 (L1/L2+Spec) show the results on the extreme UDA setup combining real and synthetic corruptions along with partial occlusion from Occluded-OOD-CV and Imagenet-C datasets. This is an extremely challenging setup where three different kinds of nuisances/domain differences are combined and is reflected in NeMo's results. Our model is still able to adapt to such a target domain, showing our method's efficacy.

Further **experimental and ablation analysis** is deferred to the Appendix due to limited space.

## 5   CONCLUSION, LIMITATIONS AND FUTURE WORK

In this work, we attempt to solve the previously unaddressed problem of **unsupervised image-only source-free domain adaptation** for 3D pose estimation. We focus our efforts on real world data with real world nuisances like weather, shape, texture, etc. and show that our method achieves significant success. Our method has limitations as it relies on the source model and cannot be trivially extended to articulated objects. It requires multiple pre-rendered samples for pose estimation. Like many other pose estimation methods, inference requires optimization using render-and-compare methodology for optimal pose estimation. In the future, we want to extend our method to unsupervised 6D pose estimation and to unseen object setups. Furthermore, the importance of the concentration parameter $\kappa$ needs more research, as we believe that it is a crucial uncertainty marker that may be relevant in more difficult domain transfer settings.

## ACKNOWLEDGEMENTS

This research has been supported by Army Research Laboratory award W911NF2320008 and ONR with N00014-21-1-2812. Adam Kortylewski acknowledges support via his Emmy Noether Research Group funded by the German Science Foundation (DFG) under Grant No. 468670075.

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

## A APPENDIX

### A.1 PROBLEM SETUP

Our problem is set up as *Source-Free, Image Only Unsupervised Domain Adaptation for Category Level 3D Pose Estimation*. We have a source 3D pose estimation model which has been trained on annotated source domain data (considered in-domain or IID). However, during adaptation, we do not have the source data. During the adaptation and evaluation time, we only have images of the OOD target domain. We don't have any 3D CAD model, LIDAR data, depth data or keypoint annotations for the OOD target domain data. Models are expected to adapt the source model in an unsupervised manner with the target images only. Our image data and pre-processing setup follow previous 3D pose estimation works like Zhou et al. (2018); Wang et al. (2021a).

**Previous Semi-supervised 3D Unsupervised Domain Adaptation Works.** There have been a few attempts in solving the domain adaptation problem in the field of 3D pose estimation; however, most domain adaptation models utilize some kind of 3D data like depth map, point cloud, or CAD models in the target domain. Some works have also tried to do semi-supervised learning using synthetic data (Yang et al., 2023; Loghmani et al., 2020) (Synthetic to Real), but in addition to requiring ground truth 3D annotations in the target domain or some target domain annotated data, they may not be applicable to real world OOD scenarios (Lee et al., 2022). Such method also seems to be partially successful since they may adapt from a specific Synthetic to Real domain but cannot trivially extend to realistic nuisance-ridden real world data.

### A.2 QUALITATIVE RESULTS

Figure 3 shows qualitative results from our method as compared to ground truth and another feature level generative model (Wang et al., 2021a) in different kinds of real world nuisances, partial occlusion and synthetic corruptions.

**Selective Vertex Adaptation and advantages w.r.t. Occlusion** The presence of partial occlusion in addition to real or synthetic nuisance factors in the target domain further complicates the already difficult UDA problem. Methods like global pseudo-labeling which utilize the entire neural mesh model will suffer immensely since now the parts of the object are individually affected - with some obscured and others visible. Our selective vertex adaptation is helpful in this complex UDA scenario, as we focus independently on the adaptation of individual neural vertices (and the model) and, as shown in our experiments 4.1 and Figure 3 performs exceedingly well even in the presence of partial occlusion in the noise-ridden target domain.

### A.3 PROOF OF THEOREM 1

We first restate the theorem,

**Theorem A.1** *A target domain $\mathcal{X}_{\mathcal{T}}$ satisfying assumption 3.3, elicits another target domain $\mathcal{X}_{\mathcal{T}}^e$ such that each sample in $\mathcal{X}_{\mathcal{T}}^e$ satisfies the global-pseudo labelling constraint ($\mathcal{L}_{sim}(f_{i \to r}, C_r) > \delta_r \, \forall \, r \in \{1, 2, ..., R\}$). Asymptotically with the size of the domain, $\mathcal{X}_{\mathcal{T}}^e \to \mathcal{X}_{\mathcal{T}}$.*

*Proof*: The CNN feature corresponding to vertex feature $C_r$ is $f_{i \to r}$. Given a vertex $K$-partition, for each subset of vertex features $I_k$, let $f^{-k}$ denote the pre-image of these vertex features. As vertex feature distributions are independent, there exists such a pre-image for each vertex feature subset. There may be uneven support in the individual pre-image sets $f^{-k}$, but the pre-image sets can be sampled such that the resulting distribution of vertex features matches $P^{I_k}$ for a given domain $\mathcal{X}$. The same pre-image argument extends to CNN features and the corresponding inputs that produce them. Let $\mathcal{X}^{-1}$ denote the pre-image of the required CNN features which elicit the distribution $P^{I_k}$ over each vertex feature subset $I_k$.

For a given target domain $\mathcal{X}_{\mathcal{T}}$ satisfying assumption 3.3 and its corresponding vertex $K$-partition, we compute the $k\delta$-subsets. Let the pre-images of vertex features for these $k\delta$-subsets be denoted by $f^{-k\delta}$ and the corresponding pre-image of the CNN features be denoted by $\mathcal{X}_{\mathcal{T}}^{-1}$. By construction, this pre-image is the target domain $\mathcal{X}_{\mathcal{T}}^e$ such that each vertex feature $C_r$ produced for samples in

this domain satisfies the global pseudo-labelling constraint. This hypothetical domain may not be easy to access but as asymptotically $\prod_k P_T^{I_k \delta}$ converges to $P_T^*$, this pre-image $\mathcal{X}_{\mathcal{T}}^{-1}$ converges to the given $\mathcal{X}_{\mathcal{T}}$ itself. □

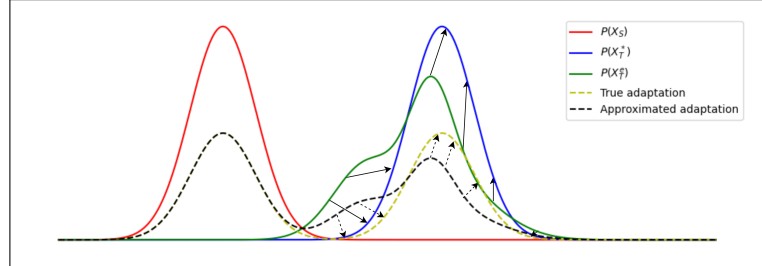

Figure 4: The elicited target distribution $P(X_T^e)$ found by SVA may not be precisely the same as the true target distribution $P(X_T^*)$, but asymptotically (shown by arrows) it tends to the true distribution and the same happens to the adapted source model.

## A.4 METHOD MOTIVATION

Figure 1 gives an intuition into our method motivation. Our method is intuitively motivated by two simple, yet crucial intuitions. *Local Part Pose Ambiguity* and *Local Part Robustness*. Local part pose ambiguity refers to the pose ambiguity that occurs when we can see only a part of an object. This is reflected in Figure 1 car object. If we could only see the right headlight (in the red box) of the car, there are many possible global poses that the car could be in. This is often the problem is 3D pose estimation, but we utilize this pose ambiguity to our advantage. We realize that since the object parts may be visible from numerous object poses, we can still update our model parameters, which correspond to these parts even when the global pose estimation is incorrect. Our second observation deals with Robust Local parts. We observe and provide evidence in Figures 1, 5, 8 and 9 of the polar histograms of some vertex features (which roughly correspond to parts of an object) being less affected or robust in the OOD data. As is trivially observed in the real world, there are some parts of objects that are less affected or even affected by certain domain changes, and we can leverage this local robustness. For example, the right headlight in Figure 1 is a part of an object which may not change much even when encountering domain nuisances like texture, weather, and context changes and will still be fairly similar to features corresponding to car headlights in the source domain.

Therefore, we are able to utilize the aforementioned observations to adapt our source model. Given local robust parts that will have high similarity to source counterparts and the fact that local parts are visible for a large number of global object poses, we can focus on these individual local parts and utilize them to update our model.

## A.5 ABLATION STUDY AND METHOD ANALYSIS

**Selective Vertex Feature Update** We can ablate the efficacy of our selective vertex feature update method and *justify our assumption regarding robust parts of an object* by comparing the (azimuth) polar histograms in Figure 5. The upper histogram represents the ratio of vertex feature activated (over visible vertex features) within a high similarity threshold on the OOD target data with a source model. It shows that even though they may not be enough robust vertex feature adaptation for a global pseudo labeling set-up, there are still *individual vertex that correspond to parts of an object which can still be detected* robustly. This is a simple observation in the real world as well, where humans identify objects in new circumstances or context by drawing upon their knowledge and recognition of individual parts which have not changed much.

**False Positives in Vertex-Feature Similarity** In our experiments, false positives were not a major issue. For example, the average ratio of false positives in Figure 5 is 2% and has an insignificant affect on the adaptation. We believe this may be to a number of reasons: (1) Our CNN backbone is trained and adapted in a contrastive manner where all unique vertex features are encouraged to be different from one another and from the background clutter features (2) While adaptation, we constrain the similarity measure with high thresholds as well require a lower bound of vertices being activated. (3) Pretrained CNN like Resnet50 have shown to have some invariance to 3D pose and other nuisances, which also contributes to the robustness of the similarity measure.

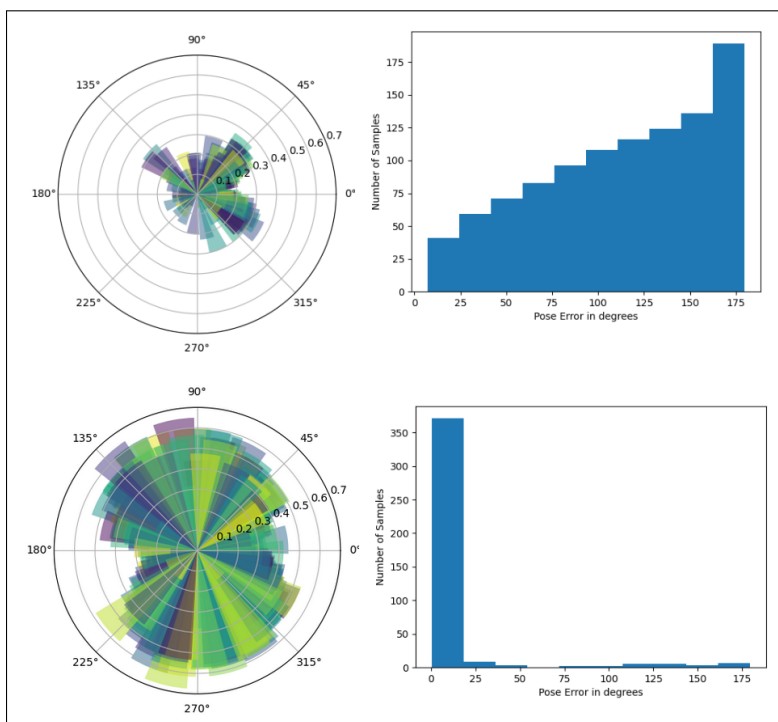

Figure 5: (Upper left) Azimuth Polar histogram represents the ratio of visible neural mesh vertices which are detected (for OOD-CV bus category), with high similarity threshold($= 0.9$) to feature vectors, by the source model. As we hypothesize in this work, the average number of robust vertices in an image varies from $5 \to 20\%$. The ratio of false positives $2\%$ per vertex feature is negligent and its effect is minimized due to the sampling of features from multiple image feature maps. (Top right) The cumulative pose error histogram for the source model evaluated on the OOD-CV (Combined Nuisance) target domain. (Lower left) Polar histogram of robust neural mesh vertices for our unsupervised adapted model which shows the major increase in adapted and robust vertex features. (Lower right) shows a histogram of pose errors for our adapted model and shows how the pose error has been minimized for the target domain.

**Multi-Pose Initialization** Figure 6 shows the estimated pose by our source model when evaluated on a sample from the OOD-CV (Zhao et al., 2023) dataset. We have three highly likely solutions learned by optimizing the reconstruction loss between the rendered feature map and the image feature map when the initial pose were randomly initialized.

**Quality of Vertex Features after Selective Adaptation** Figure 5 shows before and after results of the average ratio of correctly matched visible vertices per image in the target domain. The polar histogram is defined over the azimuth of the object pose in the target dataset. We can see significant improvements where the correctly detected ratio of vertices increases from $5 - 20\%$ to $50 - 70\%$ which is enough to estimate the pose of an object in an image. This improvement is reflected in the pose error histograms in Figure 5. Similar improvements for other classes (in OOD-CV) can be seen in Figures 8, 9.

**Ablation Analysis for Clutter Model Hyperparameter** $N$ As can be observed from Table 4, our method is not sensitive to the change in the number of clutter models. This could be attributed to the fact that our similarity threshold can be construed as a clutter model with uniform distribution. We do see small improvements as the number of clutter models is drastically increased, which comes at a high computational cost. Our default value of this hyperparameter is 5.

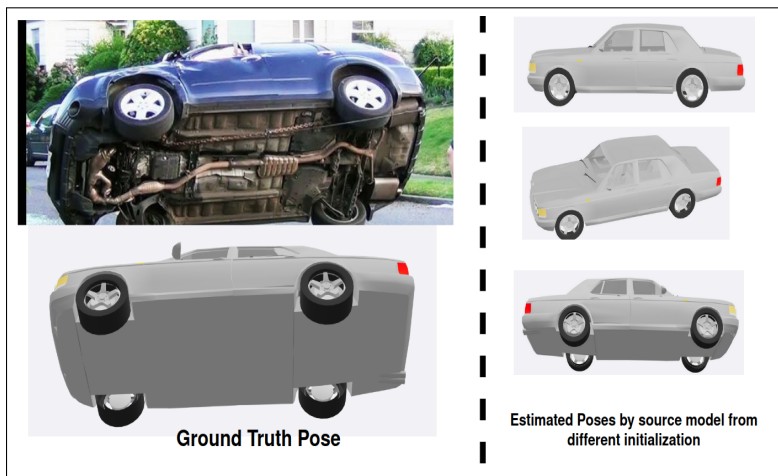

Figure 6: Neural Render-and-Compare optimized pose estimates for single image from target domain (OOD-CV) with different initialization for the source model

Table 4: Ablation Analysis of clutter model number hyperparameter

| N (Clutter) | Median Error | $\pi/18$ Acuracy | $\pi/6$ Accuracy |
|---|---|---|---|
| 1 | 3.01 | 90.1 | 95.1 |
| 5 | 3.03 | 91.1 | 94.4 |
| 20 | 2.99 | 92.3 | 95.5 |
| 35 | 2.79 | 91.7 | 94.5 |

## A.6 ADDITIONAL RELATED WORKS

Although there has not been any recent work on source free and image-only unsupervised domain adaptation for 3D pose estimation, there have been few works which have worked on self-supervised, zero-shot or few shot learning with some 3D data or multi-view availability. Goodwin et al. (2022) utilize a pretrained DINO Vision Transformer, large-scale pretraining, depth estimates and multiple views of the target object to do zero shot category level pose estimation using the idea of semantic keypoint correspondence between pairs of images. Zhang et al. (2022); He et al. (2022) and Lin et al. (2022) all utilize mesh model priors for a self-supervised approach involving 3d-2d correspondence with the mesh model or point cloud, image registration and deformation. All of these methods rely on local and global geometric constraints and correspondence to varying extents.

## A.7 TRAINING AND IMPLEMENTATION DETAILS

Our baseline model implementations are taken from their official codebase. We utilize the implementation of Ma et al. (2022) for the Resnet50-General model. The general Resnet50 model is trained with data from all categories at the same time, which could be a factor for better OOD robustness relative to other baseline models. Yang et al. (2023) is a semi-supervised setup where they create synthetic data to enable pose estimation with real data. The synthetic data creation code is not publicly available; we utilize their reported OOD-CV (Zhao et al., 2023) results in our experimental results.

During inference, it takes our method 0.33 seconds to evaluate one sample on a NVIDIA RTX 2060 GPU.

### A.7.1 SOURCE MODEL

Our source model is similar to previous works like Wang et al. (2021a; 2023); Ma et al. (2022) on category-level pose estimation using neural feature level render and compare. These methods

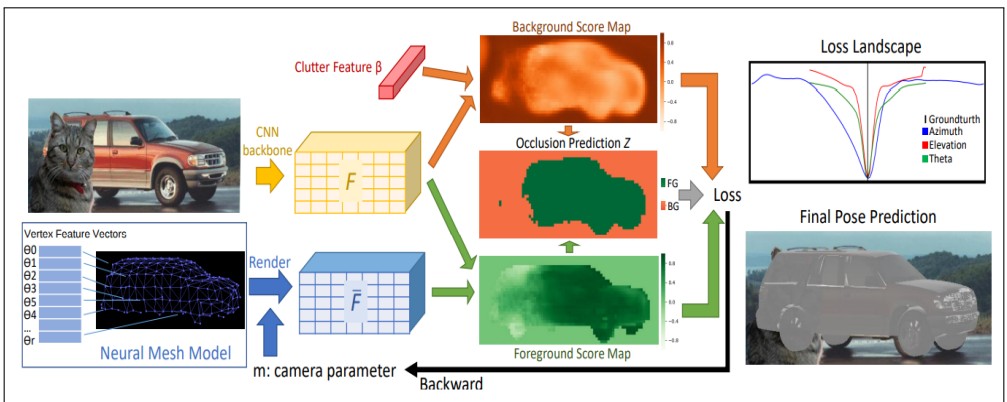

Figure 7: Source Model: Figure taken from Wang et al. (2021a) to reflect the training of our source model

are themselves 3D extensions of feature generative models such as Kortylewski et al. (2020). Our source model defines a probabilistic generative model of normalized real-valued feature activations $F$ conditioned on a 3D neural mesh representation $\mathfrak{M}$. The neural mesh model aims to capture the 3D information of the foreground objects. For each object category $y$, the source model defines a neural mesh $\mathfrak{M}_{\mathcal{S}}$ as $\{\mathcal{V}, \mathcal{C}\}$, where $\mathcal{V} = \{V_r \in \mathbb{R}^3\}_{r=1}^R$ is the set of vertices of the mesh and $\mathcal{C} = \{C_r \in \mathbb{R}^c\}_{r=1}^R$ is the set of learnable features, i.e. neural features. $r$ denotes the index of the vertices. $R$ is the total number of vertices. We also define a clutter model $\mathcal{B} = \{\beta_n\}_{n=1}^N$ to describe the backgrounds. $N$ is a prefixed hyperparameter. For a given object pose or camera viewpoint $g$, we can render the neural mesh model $\mathfrak{M}_{\mathcal{S}}$ into a feature map using (differentiable) rasterization (Kato et al., 2020). We can compute the object likelihood of a target feature map $F \in \mathcal{R}^{H \times W \times D}$ as

$$p(F|\mathfrak{M}, g, \mathcal{B}) = \prod_{i \in \mathcal{FG}} p(f_i|\mathfrak{M}, g) \prod_{i' \in \mathcal{BG}} p(f_{i'}|B), \tag{6}$$

where $\mathcal{FG}$ and $\mathcal{BG}$ denote the foreground and background pixels, respectively. $\mathcal{FG}$ is set of all the positions in the 2D lattice $P$ covered by the mesh $\mathfrak{M}$ and $\mathcal{BG}$ are the positions that are not. Unlike (Ma et al., 2022; Wang et al., 2023), we define $P(f_i|\mathfrak{M}(V_r, C_r), g) = Z[\kappa_r] \exp\{\kappa_{r,c} f_i \cdot n_{r,c}\}$ as a von Mises Fisher (vMF) distribution with mean $C_r$ and concentration parameter $\kappa_r$. $Z[\kappa_r]$ is a normalization constant and is defined as $\frac{\kappa^{d/2-1}}{(2\pi)^{d/2} \mathfrak{I}_{d/2-1}^v(\kappa)}$ (Mardia & Jupp, 2009) where $\mathfrak{I}$ denotes the modified Bessel function of the first kind of order $v$ and $d$ is the dimension. $P(f_i|b)$ is the background distribution, also defined as a vMF distribution. The correspondence between the image feature $f_i$ and the vertex features $C_r$ is calculated using rendering assuming perspective projection. The concentration parameter can be calculated using the simple approximation $\hat{\kappa} = \frac{\bar{R}(d-\bar{R}^2)}{1-\bar{R}^2}$ (Sra, 2012) where $\bar{R} = \|\bar{x}\|$ and $\bar{x} = \frac{1}{N} \sum_i^N x_i$, for a series of $N$ independent unit vectors $x$. However, for our source model, we find that fixing $\kappa$ to a constant is sufficient to learn a good supervised model. The training objective function of the source model is $\mathcal{L}_{\text{source}} = \mathcal{L}_{\text{neural}} + \mathcal{L}_{\text{clutter}} + \mathcal{L}_{\text{contrastive}}$. $\mathcal{L}_{\text{neural}}$ maximizes the likelihood between the neural features $C_r$ and the corresponding CNN features $f_{i \to r}$ for each vertex $r$. $\mathcal{L}_{\text{clutter}}$ maximizes the likelihood between the $\mathcal{B}$ and the CNN features $f_{i'}$. $\mathcal{L}_{\text{contrastive}}$ defines two constrastive terms which encourage vertex features to be different from one another and the background.

Figure 7 gives an outline of our source model which is similar to the feature-level neural render and compare methodology introduced in (Wang et al., 2021a; Ma et al., 2022). The training objective function of the source model is defined in (Wang et al., 2021a; Ma et al., 2022) as follows:

$$\mathcal{L}_{\text{source}} = \mathcal{L}_{\text{neural}} + \mathcal{L}_{\text{clutter}} + \mathcal{L}_{\text{contrastive}}. \tag{7}$$

$$\mathcal{L}_{\text{neural}} = -\sum_{r=1}^R \log P(f_{i \to r}|C_r); \quad \mathcal{L}_{\text{clutter}} = -\sum_{i' \in M} \log P(f_{i'}|\beta), \tag{8}$$

$$\mathcal{L}_{\text{contrastive}} = \lambda \sum_{r=1}^{R} \sum_{l \in R, l \notin \mathcal{N}_r} \log P(f_{i \to r}|C_l) + \mu \sum_{r=1}^{R} \sum_{n=1}^{N} \log P(f_{i \to r}|\beta_n) \qquad (9)$$

$\mathcal{L}_{\text{neural}}$ maximizes the likelihood between the neural features $C_r$ and the corresponding CNN features $f_{i \to r}$ for each vertex $r$. $\mathcal{L}_{\text{clutter}}$ maximizes the likelihood between the $\mathcal{B}$ and the CNN features $f_{i'}$. $\mathcal{L}_{\text{contrastive}}$ defines two constrastive terms which encourage vertex features to be different from one another and the background. The first term pertains to the contrastive learning between different vertices, while the second term focuses on the contrastive learning between a vertex and the background. Notably, we can do inference with the source model with only a RGB image which is one of the major reasons to utilize these class of models as our source model. However, despite earlier attempts (Yang et al., 2023; Wang et al., 2023) these models cannot be trivially extended to source-free unsupervised learning.

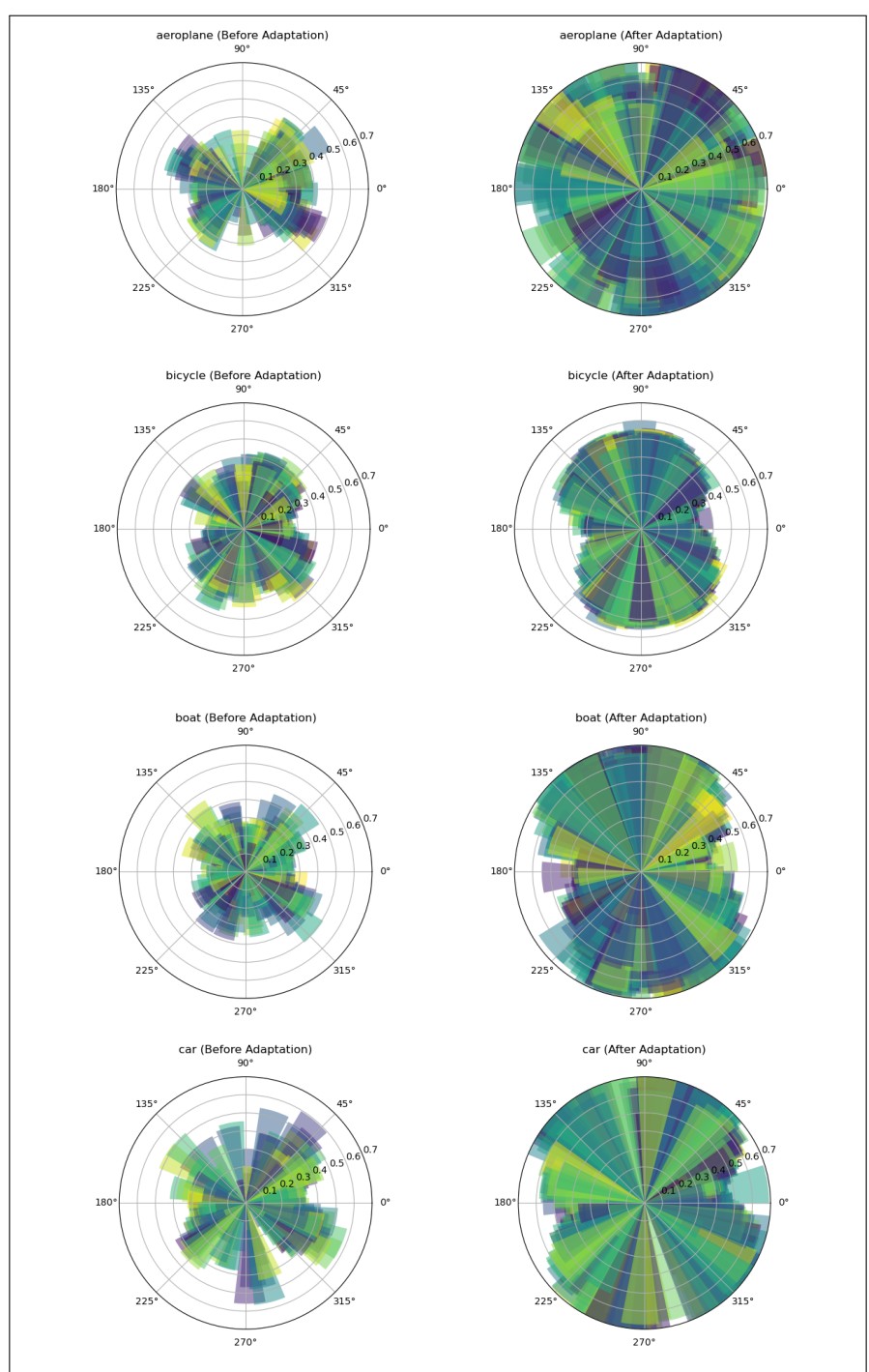

Figure 8: Part 1: Azimuth Polar histograms representing the ratio of visible neural mesh vertices which are robustly detected for different categories of OOD-CV (Zhao et al., 2023) dataset before and after adaptation using our method. As explained in 5, (left column) we can see the ratio of robustly detected vertices in the target domain using the source model which provides strong evidence towards our hypothesis regarding locally robust neural vertex features irrespective of the correctness of the global object pose. Our method, 3DUDA, leverages these locally robust parts and adapts the model in an unsupervised manner. The right column reflects the drastic increase of ratio of robustly detected vertex features post adaptation.

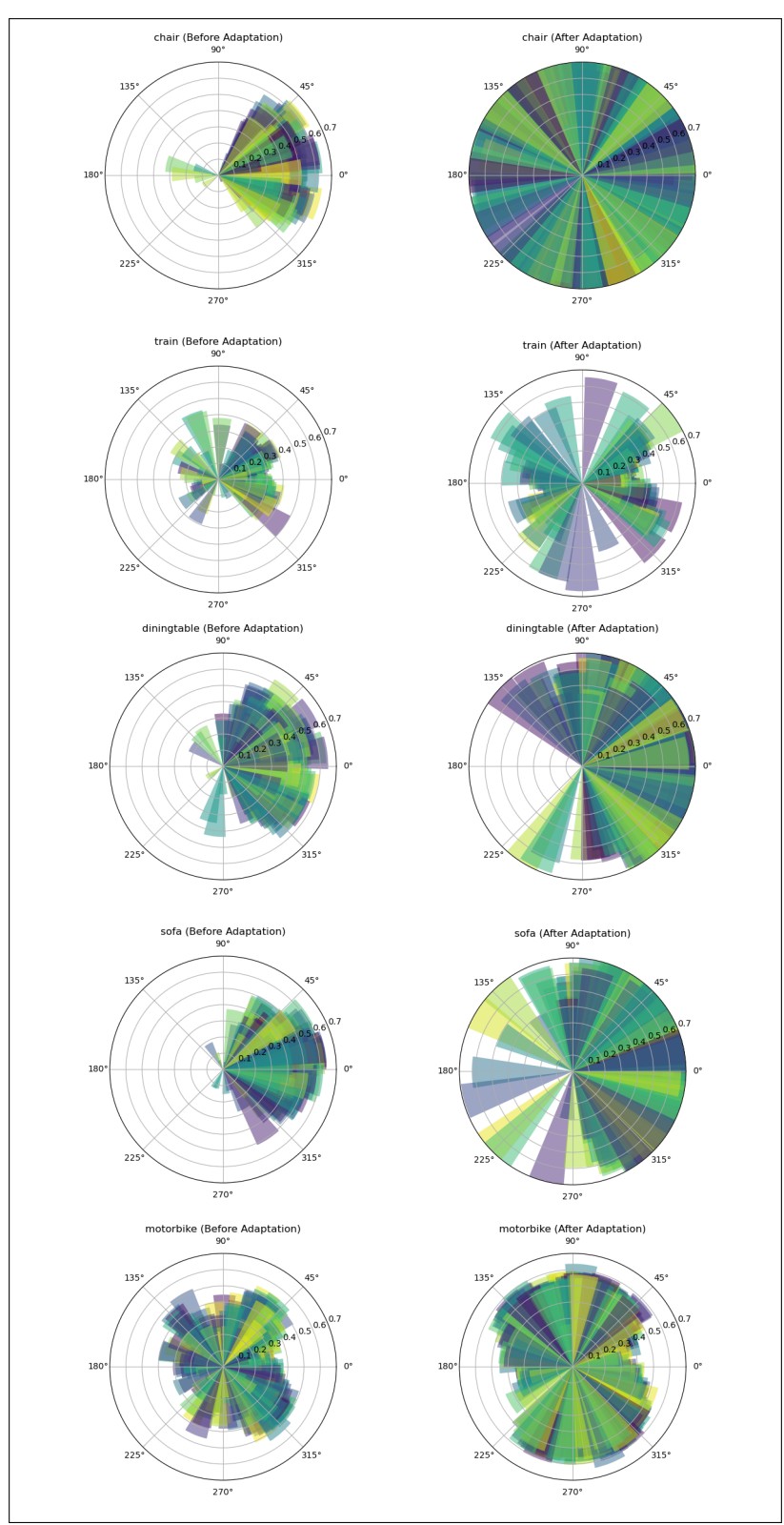

Figure 9: Part 2: Azimuth Polar histograms representing the ratio of visible neural mesh vertices which are robustly detected for different categories of OOD-CV (Zhao et al., 2023) dataset before and after adaptation using our method.

## A.8 CATEGORY-WISE EXPERIMENTAL RESULTS

We list the results of our unsupervised pose estimation results with category-wise separation in this section. Due to the sheer number of experimental results, we focus on our main comparison NeMo (Wang et al., 2021a) which has been shown to be robust to certain OOD scenarios and occlusion (Wang et al., 2023).

## A.9 EXTENDED MAIN RESULTS

Table 5: Unsupervised 3D pose estimation results for Pascal3d+ → Corrupted-Pascal3D+

(a)      (Metrics : $\pi\backslash6$ Accuracy ($\frac{\pi}{6}$), $\pi\backslash18$ Accuracy ($\frac{\pi}{18}$), Median Error (Er))

| | $\frac{\pi}{6}$↑ | $\frac{\pi}{18}$↑ | Er↓ | $\frac{\pi}{6}$↑ | $\frac{\pi}{18}$↑ | Er↓ | $\frac{\pi}{6}$↑ | $\frac{\pi}{18}$↑ | Er↓ |
|---|---|---|---|---|---|---|---|---|---|
| | **Shot Noise** | | | **Impulse Noise** | | | **Defocus Blur** | | |
| Res50-General | 39.3 | 8.9 | 52.1 | 36.6 | 8.1 | 55.7 | 45.6 | 10.3 | 44.6 |
| ViT-b-16 | 39.1 | 8.8 | 55.4 | 36.7 | 8.4 | 54.9 | 45.2 | 13.1 | 41.9 |
| DMNT | 51.2 | 21.1 | 31.1 | 50.1 | 25.3 | 33.3 | 72.1 | 44.5 | 16.1 |
| P3D | 48.9 | 19.7 | 33.7 | 48.6 | 18.9 | 37.9 | 71.6 | 40.7 | 17.8 |
| NeMo | 50.6 | 25.3 | 35.0 | 45.4 | 22.2 | 39.4 | 72.9 | 41.8 | 16.0 |
| Ours | **85.9** | **62.0** | **9.0** | **84.0** | **58.5** | **10.1** | **87.8** | **64.6** | **8.0** |
| | **Motion Blur** | | | **Zoom Blur** | | | **Snow** | | |
| Res50-General | 42.3 | 9.2 | 49.9 | 59.6 | 18.4 | 41.0 | 55.4 | 16.2 | 40.4 |
| ViT-b-16 | 46.1 | 8.7 | 49.8 | 53.9 | 12.1 | 40.7 | 54.3 | 12.8 | 40.3 |
| DMNT | 70.7 | 41.1 | 16.9 | 70.9 | 42.1 | 16.6 | 72.2 | 44.6 | 16.6 |
| P3D | 69.9 | 37.6 | 17.9 | 70.0 | 40.0 | 18.6 | 72.0 | 41.9 | 17.8 |
| NeMo | 69.7 | 39.2 | 18.7 | 69.0 | 39.7 | 19.1 | 69.9 | 40.1 | 18.9 |
| Ours | **88.0** | **63.8** | **8.3** | **87.9** | **65.1** | **8.1** | **87.7** | **64.0** | **8.2** |
| | **Fog** | | | **Contrast** | | | **Elastic Transform** | | |
| Res50-General | 65.3 | 19.6 | 31.5 | 61.5 | 19.3 | 38.1 | 45.6 | 8.6 | 43.7 |
| ViT-b-16 | 65.7 | 21.8 | 29.9 | 61.9 | 21.5 | 37.7 | 41.3 | 6.3 | 47.9 |
| DMNT | 86.0 | 60.1 | 9.7 | 77.7 | 45.9 | 12.9 | 77.9 | 51.1 | 13.7 |
| P3D | 85.9 | 58.4 | 9.9 | 76.5 | 44.8 | 13.6 | 76.1 | 48.9 | 14.3 |
| NeMo | 85.5 | 59.0 | 9.5 | 74.5 | 43.8 | 14.7 | 77.4 | 50.3 | 13.8 |
| Ours | **88.7** | **65.6** | **7.8** | **88.8** | **66.7** | **7.6** | **88.2** | **64.4** | **8.1** |

### A.9.1 OOD-CV (ZHAO ET AL., 2023)

Tables 7 8 15 16 9 10 11 12 13 14 17 18 show per category results for Unsupervised Domain Adaptation for the OOD-CV (Zhao et al., 2023) dataset, where we have real world nuisances in the target domain.

### A.9.2 PASCAL3D+ → CORRUPTED-CORRUPTED PASCAL3D+

Tables 19-50 show category wise results of our Pascal3D+ to Corrupted Pascal3D+ setup where we add synthetic noise like glass blur, elastic transform, rain, fog, gaussian noise, etc. to the evaluation data.

Table 6: Unsupervised 3D pose estimation results for Occlusion and Extreme UDA setup

(a) **OccL1/L2**: Real Nuisance (OOD-CV (Combined)) + Occlusion (Level1/Level2) (b) **OOD+SN/GB**: Real Nuisance (OOD-CV) + Synthetic Noise (Speckle Noise/Glass Blur) (c) **L1/L2+Speckle**: Real Nuisance (OOD-CV) + Occlusion (L1/L2) + Synthetic Noise (Speckle Noise)

| | OccL1 | | OccL2 | | OOD+SN | |
|---|---|---|---|---|---|---|
| | $\frac{\pi}{6}$Acc. | $\frac{\pi}{18}$Acc. | $\frac{\pi}{6}$Acc. | $\frac{\pi}{18}$Acc. | $\frac{\pi}{6}$Acc. | $\frac{\pi}{18}$Acc. |
| Res50-General | 30.1 | 10.9 | 18.9 | 4.9 | 30.1 | 8.8 |
| MaskRCNN | 28.9 | 9.8 | 17.7 | 4.5 | 31.2 | 9.7 |
| DMNT | 32.4 | 11.9 | 25.3 | 7.1 | 35.4 | 10.2 |
| P3D | 29.6 | 11.4 | 22.1 | 6.9 | 33.1 | 11.1 |
| NeMo | 30.6 | 10.2 | 24.1 | 6.6 | 32.7 | 10.2 |
| Ours | **84.6** | **77.1** | **78.7** | **70.4** | **80.5** | **63.0** |
| | OOD+GB | | L1+Speckle | | L2+Speckle | |
| Res50-General | 30.3 | 10.1 | 16.7 | 3.3 | 11.8 | 1.5 |
| MaskRCNN | 28.9 | 9.9 | 15.4 | 3.1 | 12.5 | 1.8 |
| DMNT | 32.3 | 10.2 | 18.9 | 3.9 | 16.8 | 3.1 |
| P3D | 30.1 | 9.3 | 17.7 | 3.4 | 15.9 | 2.6 |
| NeMo | 29.6 | 9.5 | 18.6 | 3.4 | 15.1 | 2.7 |
| Ours | **77.7** | **65.9** | **69.4** | **50.4** | **60.6** | **38.9** |

Table 7: Unsupervised 3D pose estimation results for OOD-CV (combined) Part 1

| Method | Metric | aeroplane | bicycle | boat | bus | car |
|---|---|---|---|---|---|---|
| Wang et al. (2021a) | $\pi\backslash6$ Acc. | 45.6 | 48.5 | 39.2 | 66.0 | 68.8 |
| | $\pi\backslash18$ Acc. | 25.0 | 20.8 | 14.7 | 47.4 | 52.7 |
| | Median Error | 36.35 | 31.71 | 48.0 | 11.6 | 9.22 |
| Ours | $\pi\backslash6$ Acc. | 98.6 | 90.5 | 86.8 | 92.4 | 97.1 |
| | $\pi\backslash18$ Acc. | 94.3 | 88.2 | 75.0 | 88.0 | 92.9 |
| | Median Error | 1.45 | 1.44 | 3.6 | 1.75 | 1.82 |

Table 8: Unsupervised 3D pose estimation results for OOD-CV (combined) Part 2

| Method | Metric | chair | diningtable | motorbike | sofa | train |
|---|---|---|---|---|---|---|
| Wang et al. (2021a) | $\pi\backslash6$ Acc. | 42.6 | 57.6 | 57.2 | 73.2 | 77.8 |
| | $\pi\backslash18$ Acc. | 14.8 | 19.0 | 23.5 | 26.1 | 60.5 |
| | Median Error | 42.63 | 23.65 | 24.16 | 17.89 | 7.06 |
| Ours | $\pi\backslash6$ Acc. | 93.0 | 96.5 | 96.0 | 97.7 | 92.9 |
| | $\pi\backslash18$ Acc. | 80.8 | 88.2 | 89.2 | 91.6 | 89.5 |
| | Median Error | 3.3 | 1.82 | 1.73 | 1.7 | 2.59 |

Table 9: Unsupervised 3D pose estimation results for OOD-CV (shape) Part 1

| Method | Metric | aeroplane | bicycle | boat | bus | car |
|---|---|---|---|---|---|---|
| Wang et al. (2021a) | $\pi\backslash6$ Acc. | 41.4 | 58.6 | 52.8 | 76.7 | 76.9 |
| | $\pi\backslash18$ Acc. | 14.1 | 36.9 | 13.9 | 33.3 | 50.0 |
| | Median Error | 36.0 | 15.77 | 26.34 | 17.08 | 10.95 |
| Ours | $\pi\backslash6$ Acc. | 96.1 | 91.9 | 94.4 | 90.0 | 100.0 |
| | $\pi\backslash18$ Acc. | 92.2 | 91.0 | 83.3 | 83.3 | 84.6 |
| | Median Error | 1.72 | 1.43 | 2.35 | 2.34 | 3.21 |

Table 10: Unsupervised 3D pose estimation results for OOD-CV (shape) Part 2

| Method | Metric | chair | diningtable | motorbike | sofa | train |
|---|---|---|---|---|---|---|
| Wang et al. (2021a) | $\pi\backslash6$ Acc. | 48.3 | 52.8 | 51.2 | 78.9 | 50.0 |
| | $\pi\backslash18$ Acc. | 15.2 | 12.4 | 25.6 | 26.3 | 22.2 |
| | Median Error | 33.59 | 27.61 | 28.87 | 16.31 | 26.83 |
| Ours | $\pi\backslash6$ Acc. | 92.7 | 93.3 | 93.0 | 96.8 | 77.8 |
| | $\pi\backslash18$ Acc. | 74.8 | 70.8 | 76.7 | 83.2 | 72.2 |
| | Median Error | 3.58 | 5.62 | 2.95 | 2.42 | 3.73 |

Table 11: Unsupervised 3D pose estimation results for OOD-CV (weather) Part 1

| Method | Metric | aeroplane | boat | bus | car |
|---|---|---|---|---|---|
| Wang et al. (2021a) | $\pi\backslash6$ Acc. | 54.0 | 45.8 | 64.9 | 68.6 |
| | $\pi\backslash18$ Acc. | 38.8 | 22.3 | 50.6 | 56.5 |
| | Median Error | 21.52 | 35.39 | 8.5 | 7.68 |
| Ours | $\pi\backslash6$ Acc. | 99.3 | 85.7 | 97.4 | 98.8 |
| | $\pi\backslash18$ Acc. | 96.9 | 76.9 | 89.6 | 96.6 |
| | Median Error | 2.48 | 2.29 | 1.35 | 1.49 |

Table 12: Unsupervised 3D pose estimation results for OOD-CV (weather) Part 2

| Method | Metric | chair | motorbike | sofa | train |
|---|---|---|---|---|---|
| Wang et al. (2021a) | $\pi\backslash6$ Acc. | 44.1 | 63.9 | 87.5 | 81.4 |
| | $\pi\backslash18$ Acc. | 17.6 | 23.9 | 62.5 | 68.3 |
| | Median Error | 48.9 | 20.68 | 5.42 | 5.62 |
| Ours | $\pi\backslash6$ Acc. | 97.1 | 97.8 | 100.0 | 89.8 |
| | $\pi\backslash18$ Acc. | 91.2 | 94.4 | 87.5 | 85.6 |
| | Median Error | 2.7 | 1.65 | 1.62 | 1.51 |

Table 13: Unsupervised 3D pose estimation results for OOD-CV (texture) Part 1

| Method | Metric | aeroplane | bicycle | boat | bus | car |
|---|---|---|---|---|---|---|
| Wang et al. (2021a) | $\pi\backslash6$ Acc. | 54.5 | 57.1 | 45.3 | 76.9 | 77.1 |
| | $\pi\backslash18$ Acc. | 28.7 | 13.1 | 14.0 | 57.9 | 56.2 |
| | Median Error | 24.3 | 24.15 | 44.11 | 8.01 | 7.94 |
| Ours | $\pi\backslash6$ Acc. | 99.0 | 97.6 | 88.4 | 98.3 | 100.0 |
| | $\pi\backslash18$ Acc. | 98.0 | 94.0 | 76.7 | 95.9 | 97.9 |
| | Median Error | 1.68 | 1.68 | 2.75 | 1.68 | 2.61 |

Table 14: Unsupervised 3D pose estimation results for OOD-CV (texture) Part 2

| Method | Metric | chair | diningtable | motorbike | sofa | train |
|---|---|---|---|---|---|---|
| Wang et al. (2021a) | $\pi\backslash6$ Acc. | 36.2 | 65.4 | 59.3 | 67.0 | 72.3 |
| | $\pi\backslash18$ Acc. | 12.1 | 24.1 | 28.7 | 25.3 | 56.3 |
| | Median Error | 62.55 | 18.07 | 22.35 | 18.77 | 7.52 |
| Ours | $\pi\backslash6$ Acc. | 94.8 | 99.0 | 97.2 | 96.7 | 97.5 |
| | $\pi\backslash18$ Acc. | 86.2 | 94.8 | 92.6 | 92.3 | 95.8 |
| | Median Error | 3.01 | 1.66 | 1.99 | 2.16 | 1.61 |

Table 15: Unsupervised 3D pose estimation results for OOD-CV (context) Part 1

| Method | Metric | aeroplane | boat | bus | car |
|---|---|---|---|---|---|
| Wang et al. (2021a) | $\pi\backslash6$ Acc. | 34.1 | 34.3 | 65.4 | 69.4 |
| | $\pi\backslash18$ Acc. | 12.1 | 10.1 | 50.7 | 45.8 |
| | Median Error | 47.11 | 49.27 | 9.61 | 11.38 |
| Ours | $\pi\backslash6$ Acc. | 98.8 | 89.9 | 91.9 | 97.2 |
| | $\pi\backslash18$ Acc. | 94.8 | 76.9 | 87.5 | 91.7 |
| | Median Error | 1.61 | 3.64 | 1.67 | 2.41 |

Table 16: Unsupervised 3D pose estimation results for OOD-CV (context) Part 2

| Method | Metric | chair | diningtable | motorbike | sofa | train |
|---|---|---|---|---|---|---|
| Wang et al. (2021a) | $\pi\backslash6$ Acc. | 43.7 | 49.0 | 55.2 | 73.1 | 84.9 |
| | $\pi\backslash18$ Acc. | 18.3 | 18.3 | 21.8 | 25.4 | 57.0 |
| | Median Error | 35.34 | 30.4 | 23.36 | 18.5 | 7.52 |
| Ours | $\pi\backslash6$ Acc. | 94.4 | 98.1 | 94.3 | 99.5 | 94.2 |
| | $\pi\backslash18$ Acc. | 90.1 | 90.4 | 82.8 | 96.4 | 91.9 |
| | Median Error | 3.62 | 2.12 | 1.9 | 1.56 | 1.57 |

Table 17: Unsupervised 3D pose estimation results for OOD-CV (pose) Part 1

| Method | Metric | aeroplane | bicycle | boat | bus | car |
|---|---|---|---|---|---|---|
| Wang et al. (2021a) | $\pi\backslash6$ Acc. | 7.1 | 15.1 | 12.7 | 33.3 | 53.8 |
| | $\pi\backslash18$ Acc. | 0.0 | 1.6 | 0.0 | 11.1 | 32.7 |
| | Median Error | 107.37 | 81.41 | 113.72 | 48.94 | 17.36 |
| Ours | $\pi\backslash6$ Acc. | 96.4 | 73.8 | 79.4 | 80.0 | 92.3 |
| | $\pi\backslash18$ Acc. | 64.3 | 69.8 | 58.7 | 71.1 | 71.2 |
| | Median Error | 8.74 | 3.59 | 7.39 | 3.01 | 3.1 |

Table 18: Unsupervised 3D pose estimation results for OOD-CV (pose) Part 2

| Method | Metric | chair | diningtable | motorbike | sofa | train |
|---|---|---|---|---|---|---|
| Wang et al. (2021a) | $\pi\backslash6$ Acc. | 23.1 | 20.0 | 25.0 | 0.0 | 65.0 |
| | $\pi\backslash18$ Acc. | 7.7 | 0.0 | 7.1 | 0.0 | 45.0 |
| | Median Error | 119.05 | 66.66 | 76.21 | 88.57 | 10.8 |
| Ours | $\pi\backslash6$ Acc. | 76.9 | 80.0 | 100.0 | 100.0 | 95.0 |
| | $\pi\backslash18$ Acc. | 53.8 | 80.0 | 78.6 | 100.0 | 90.0 |
| | Median Error | 6.54 | 3.55 | 2.89 | 1.49 | 2.6 |

Table 19: Unsupervised 3D pose estimation for Corrupted-Pascal3D+ (gaussian noise) Part 1

| Method | Metric | plane | bicycle | boat | bottle | bus | car |
|--------|--------|-------|---------|------|--------|-----|-----|
| (Wang et al., 2021a) | $\frac{\pi}{6}$ Accuracy | 16.7 | 29.9 | 13.1 | 17.1 | 47.2 | 80.3 |
| | $\frac{\pi}{18}$ Accuracy | 1.7 | 3.3 | 1.9 | 0.7 | 23.5 | 64.0 |
| | Median Error | 84.23 | 60.37 | 86.0 | 45.71 | 37.68 | 7.4 |
| **Ours** | $\frac{\pi}{6}$ Accuracy | 79.8 | 71.9 | 59.9 | 86.9 | 95.7 | 97.9 |
| | $\frac{\pi}{18}$ Accuracy | 46.3 | 16.6 | 32.1 | 48.1 | 91.0 | 93.8 |
| | Median Error | 11.04 | 20.2 | 20.08 | 10.43 | 3.22 | 3.47 |

Table 20: Unsupervised 3D pose estimation for Corrupted-Pascal3D+ (gaussian noise) Part 2

| Method | Metric | chair | table | bike | sofa | train | tv |
|--------|--------|-------|-------|------|------|-------|-----|
| (Wang et al., 2021a) | $\frac{\pi}{6}$ Accuracy | 42.3 | 28.3 | 31.3 | 34.5 | 46.6 | 69.1 |
| | $\frac{\pi}{18}$ Accuracy | 8.5 | 2.5 | 3.9 | 6.4 | 18.6 | 19.8 |
| | Median Error | 35.04 | 48.16 | 49.55 | 50.1 | 34.37 | 21.17 |
| **Ours** | $\frac{\pi}{6}$ Accuracy | 85.8 | 78.1 | 77.6 | 89.9 | 89.0 | 79.1 |
| | $\frac{\pi}{18}$ Accuracy | 50.4 | 55.9 | 23.6 | 53.7 | 75.5 | 39.3 |
| | Median Error | 9.92 | 8.63 | 17.16 | 9.22 | 5.44 | 12.86 |

Table 21: Unsupervised 3D pose estimation results for Corrupted-Pascal3D+ (shot noise) Part 1

| Method | Metric | plane | bicycle | boat | bottle | bus | car |
|--------|--------|-------|---------|------|--------|-----|-----|
| (Wang et al., 2021a) | $\frac{\pi}{6}$ Accuracy | 23.4 | 33.5 | 22.0 | 19.1 | 62.6 | 86.9 |
| | $\frac{\pi}{18}$ Accuracy | 5.1 | 5.3 | 3.5 | 2.0 | 37.4 | 71.4 |
| | Median Error | 69.94 | 49.79 | 75.4 | 47.59 | 15.39 | 6.91 |
| **Ours** | $\frac{\pi}{6}$ Accuracy | 79.6 | 74.9 | 63.5 | 87.3 | 94.9 | 98.2 |
| | $\frac{\pi}{18}$ Accuracy | 49.7 | 23.6 | 35.5 | 48.9 | 90.8 | 94.3 |
| | Median Error | 10.1 | 17.4 | 17.98 | 10.61 | 2.94 | 3.33 |

Table 22: Unsupervised 3D pose estimation results for Corrupted-Pascal3D+ (shot noise) Part 2

| Method | Metric | chair | table | bike | sofa | train | tv |
|---|---|---|---|---|---|---|---|
| (Wang et al., 2021a) | $\frac{\pi}{6}$ Accuracy | 42.9 | 39.1 | 39.9 | 43.1 | 53.4 | 71.5 |
| | $\frac{\pi}{18}$ Accuracy | 10.1 | 6.3 | 3.9 | 8.5 | 21.7 | 19.5 |
| | Median Error | 35.53 | 39.29 | 37.74 | 34.38 | 25.59 | 20.39 |
| **Ours** | $\frac{\pi}{6}$ Accuracy | 88.1 | 80.6 | 83.7 | 92.0 | 89.8 | 82.0 |
| | $\frac{\pi}{18}$ Accuracy | 53.4 | 62.3 | 32.2 | 54.8 | 77.6 | 42.0 |
| | Median Error | 9.53 | 7.35 | 14.39 | 9.06 | 5.22 | 12.28 |

Table 23: Unsupervised 3D pose estimation results for Corrupted-Pascal3D+ (impulse noise) Part 1

| Method | Metric | plane | bicycle | boat | bottle | bus | car |
|---|---|---|---|---|---|---|---|
| (Wang et al., 2021a) | $\frac{\pi}{6}$ Accuracy | 17.6 | 32.7 | 12.9 | 21.7 | 50.6 | 82.5 |
| | $\frac{\pi}{18}$ Accuracy | 2.4 | 3.6 | 1.6 | 1.1 | 25.8 | 66.9 |
| | Median Error | 80.9 | 50.67 | 83.49 | 49.34 | 29.27 | 6.94 |
| **Ours** | $\frac{\pi}{6}$ Accuracy | 80.7 | 73.0 | 57.4 | 85.1 | 93.4 | 97.7 |
| | $\frac{\pi}{18}$ Accuracy | 46.7 | 19.1 | 29.2 | 44.6 | 88.0 | 93.1 |
| | Median Error | 11.05 | 20.03 | 21.96 | 11.35 | 3.25 | 3.46 |

Table 24: Unsupervised 3D pose estimation results for Corrupted-Pascal3D+ (impulse noise) Part 2

| Method | Metric | chair | table | bike | sofa | train | tv |
|---|---|---|---|---|---|---|---|
| (Wang et al., 2021a) | $\frac{\pi}{6}$ Accuracy | 40.3 | 25.9 | 30.8 | 43.1 | 51.2 | 69.2 |
| | $\frac{\pi}{18}$ Accuracy | 7.5 | 2.2 | 3.7 | 8.0 | 19.9 | 18.0 |
| | Median Error | 37.0 | 47.17 | 48.24 | 36.48 | 28.19 | 20.42 |
| **Ours** | $\frac{\pi}{6}$ Accuracy | 88.5 | 76.8 | 79.0 | 89.1 | 88.4 | 81.0 |
| | $\frac{\pi}{18}$ Accuracy | 53.0 | 55.3 | 24.2 | 52.2 | 74.4 | 40.4 |
| | Median Error | 9.37 | 8.96 | 16.51 | 9.6 | 5.44 | 12.82 |

Table 25: Unsupervised 3D pose estimation results for Corrupted-Pascal3D+ (defocus blur) Part 1

| Method | Metric | plane | bicycle | boat | bottle | bus | car |
|---|---|---|---|---|---|---|---|
| (Wang et al., 2021a) | $\frac{\pi}{6}$ Accuracy | 52.4 | 60.9 | 44.6 | 83.0 | 69.2 | 89.7 |
| | $\frac{\pi}{18}$ Accuracy | 15.9 | 16.9 | 11.1 | 40.7 | 46.8 | 74.9 |
| | Median Error | 28.41 | 22.27 | 35.9 | 12.14 | 11.25 | 6.57 |
| **Ours** | $\frac{\pi}{6}$ Accuracy | 84.8 | 81.6 | 71.8 | 88.4 | 94.5 | 98.6 |
| | $\frac{\pi}{18}$ Accuracy | 51.9 | 31.8 | 43.3 | 53.0 | 91.5 | 94.6 |
| | Median Error | 9.67 | 15.72 | 12.11 | 9.52 | 3.06 | 3.18 |

Table 26: Unsupervised 3D pose estimation results for Corrupted-Pascal3D+ (defocus blur) Part 2

| Method | Metric | chair | table | bike | sofa | train | tv |
|---|---|---|---|---|---|---|---|
| (Wang et al., 2021a) | $\frac{\pi}{6}$ Accuracy | 82.6 | 71.6 | 64.0 | 91.3 | 57.6 | 82.9 |
| | $\frac{\pi}{18}$ Accuracy | 43.5 | 44.4 | 14.6 | 46.3 | 33.7 | 37.0 |
| | Median Error | 10.99 | 11.98 | 22.52 | 10.73 | 18.51 | 13.62 |
| **Ours** | $\frac{\pi}{6}$ Accuracy | 89.7 | 79.5 | 84.3 | 94.4 | 89.1 | 83.8 |
| | $\frac{\pi}{18}$ Accuracy | 60.3 | 62.3 | 32.2 | 61.4 | 78.1 | 43.3 |
| | Median Error | 8.63 | 7.24 | 13.89 | 8.12 | 4.9 | 11.82 |

Table 27: Unsupervised 3D pose estimation results for Corrupted-Pascal3D+ (glass blur) Part 1

| Method | Metric | plane | bicycle | boat | bottle | bus | car |
|---|---|---|---|---|---|---|---|
| (Wang et al., 2021a) | $\frac{\pi}{6}$ Accuracy | 24.2 | 31.9 | 28.5 | 78.2 | 53.6 | 75.8 |
| | $\frac{\pi}{18}$ Accuracy | 4.1 | 4.0 | 6.0 | 31.1 | 22.0 | 52.4 |
| | Median Error | 63.83 | 54.77 | 59.95 | 15.43 | 25.27 | 9.43 |
| **Ours** | $\frac{\pi}{6}$ Accuracy | 80.4 | 75.7 | 69.0 | 88.2 | 93.6 | 97.7 |
| | $\frac{\pi}{18}$ Accuracy | 47.7 | 24.3 | 39.2 | 49.1 | 89.1 | 93.8 |
| | Median Error | 10.67 | 17.37 | 14.06 | 10.24 | 3.34 | 3.39 |

Table 28: Unsupervised 3D pose estimation results for Corrupted-Pascal3D+ (glass blur) Part 2

| Method | Metric | chair | table | bike | sofa | train | tv |
|---|---|---|---|---|---|---|---|
| (Wang et al., 2021a) | $\frac{\pi}{6}$ Accuracy | 70.0 | 57.2 | 30.8 | 87.2 | 36.0 | 78.7 |
| | $\frac{\pi}{18}$ Accuracy | 29.8 | 28.5 | 4.2 | 42.5 | 14.4 | 24.2 |
| | Median Error | 15.75 | 21.99 | 46.19 | 11.66 | 116.73 | 17.54 |
| **Ours** | $\frac{\pi}{6}$ Accuracy | 88.7 | 84.0 | 81.8 | 95.2 | 86.8 | 82.3 |
| | $\frac{\pi}{18}$ Accuracy | 58.1 | 63.7 | 29.6 | 62.8 | 71.0 | 44.1 |
| | Median Error | 8.86 | 7.26 | 15.76 | 8.17 | 5.75 | 11.79 |

Table 29: Unsupervised 3D pose estimation results for Corrupted-Pascal3D+ (motion blur) Part 1

| Method | Metric | plane | bicycle | boat | bottle | bus | car |
|---|---|---|---|---|---|---|---|
| (Wang et al., 2021a) | $\frac{\pi}{6}$ Accuracy | 54.6 | 44.7 | 39.0 | 77.4 | 69.9 | 92.4 |
| | $\frac{\pi}{18}$ Accuracy | 18.6 | 10.2 | 10.4 | 19.8 | 40.6 | 78.5 |
| | Median Error | 26.85 | 35.07 | 41.92 | 17.82 | 13.07 | 5.73 |
| **Ours** | $\frac{\pi}{6}$ Accuracy | 85.1 | 80.2 | 71.3 | 87.4 | 94.5 | 98.7 |
| | $\frac{\pi}{18}$ Accuracy | 52.6 | 29.1 | 43.9 | 46.3 | 90.6 | 94.8 |
| | Median Error | 9.43 | 16.32 | 12.26 | 11.02 | 3.03 | 3.34 |

Table 30: Unsupervised 3D pose estimation results for Corrupted-Pascal3D+ (motion blur) Part 2

| Method | Metric | chair | table | bike | sofa | train | tv |
|---|---|---|---|---|---|---|---|
| (Wang et al., 2021a) | $\frac{\pi}{6}$ Accuracy | 80.2 | 64.4 | 42.1 | 90.4 | 59.8 | 79.7 |
| | $\frac{\pi}{18}$ Accuracy | 40.7 | 35.0 | 7.2 | 41.7 | 39.0 | 35.3 |
| | Median Error | 12.08 | 15.89 | 37.85 | 11.36 | 16.12 | 14.19 |
| **Ours** | $\frac{\pi}{6}$ Accuracy | 89.5 | 82.5 | 82.0 | 94.9 | 90.2 | 82.9 |
| | $\frac{\pi}{18}$ Accuracy | 53.8 | 65.3 | 29.3 | 59.8 | 77.6 | 42.2 |
| | Median Error | 9.24 | 7.32 | 15.13 | 8.26 | 5.14 | 11.74 |

Table 31: Unsupervised 3D pose estimation results for Corrupted-Pascal3D+ (zoom blur) Part 1

| Method | Metric | plane | bicycle | boat | bottle | bus | car |
|---|---|---|---|---|---|---|---|
| (Wang et al., 2021a) | $\frac{\pi}{6}$ Accuracy | 58.9 | 40.3 | 40.6 | 74.8 | 70.7 | 92.0 |
| | $\frac{\pi}{18}$ Accuracy | 22.8 | 10.1 | 16.2 | 27.3 | 45.3 | 76.9 |
| | Median Error | 22.51 | 40.54 | 42.25 | 16.8 | 11.24 | 6.15 |
| **Ours** | $\frac{\pi}{6}$ Accuracy | 86.1 | 80.2 | 66.6 | 87.8 | 94.0 | 98.5 |
| | $\frac{\pi}{18}$ Accuracy | 57.1 | 29.3 | 40.2 | 54.2 | 91.5 | 94.8 |
| | Median Error | 8.56 | 16.52 | 13.6 | 9.21 | 2.97 | 3.25 |

Table 32: Unsupervised 3D pose estimation results for Corrupted-Pascal3D+ (zoom blur) Part 2

| Method | Metric | chair | table | bike | sofa | train | tv |
|---|---|---|---|---|---|---|---|
| (Wang et al., 2021a) | $\frac{\pi}{6}$ Accuracy | 78.9 | 62.5 | 36.0 | 86.7 | 63.0 | 77.3 |
| | $\frac{\pi}{18}$ Accuracy | 37.2 | 36.0 | 7.1 | 38.9 | 38.4 | 26.6 |
| | Median Error | 12.59 | 17.87 | 41.47 | 12.17 | 14.46 | 16.68 |
| **Ours** | $\frac{\pi}{6}$ Accuracy | 91.3 | 84.0 | 82.8 | 94.6 | 91.8 | 82.3 |
| | $\frac{\pi}{18}$ Accuracy | 58.5 | 66.3 | 30.1 | 59.9 | 80.6 | 43.0 |
| | Median Error | 8.55 | 6.84 | 15.3 | 8.02 | 5.01 | 12.09 |

Table 33: Unsupervised 3D pose estimation results for Corrupted-Pascal3D+ (snow) Part 1

| Method | Metric | plane | bicycle | boat | bottle | bus | car |
|---|---|---|---|---|---|---|---|
| (Wang et al., 2021a) | $\frac{\pi}{6}$ Accuracy | 52.6 | 61.9 | 34.7 | 70.0 | 82.3 | 94.6 |
| | $\frac{\pi}{18}$ Accuracy | 20.4 | 12.9 | 10.1 | 21.8 | 63.9 | 83.4 |
| | Median Error | 27.74 | 24.37 | 51.79 | 18.92 | 7.01 | 5.33 |
| **Ours** | $\frac{\pi}{6}$ Accuracy | 86.6 | 80.9 | 69.0 | 87.1 | 94.9 | 98.7 |
| | $\frac{\pi}{18}$ Accuracy | 55.3 | 29.9 | 40.4 | 47.3 | 91.2 | 95.0 |
| | Median Error | 8.97 | 16.29 | 13.45 | 10.62 | 2.73 | 3.19 |

Table 34: Unsupervised 3D pose estimation results for Corrupted-Pascal3D+ (snow) Part 2

| Method | Metric | chair | table | bike | sofa | train | tv |
|---|---|---|---|---|---|---|---|
| (Wang et al., 2021a) | $\frac{\pi}{6}$ Accuracy | 72.3 | 58.3 | 57.4 | 72.3 | 70.7 | 75.2 |
| | $\frac{\pi}{18}$ Accuracy | 26.7 | 27.1 | 14.0 | 28.5 | 50.5 | 23.7 |
| | Median Error | 16.36 | 21.16 | 25.11 | 15.83 | 9.71 | 17.64 |
| **Ours** | $\frac{\pi}{6}$ Accuracy | 90.1 | 81.5 | 84.0 | 92.9 | 88.8 | 82.0 |
| | $\frac{\pi}{18}$ Accuracy | 55.3 | 62.4 | 34.7 | 58.2 | 77.3 | 44.6 |
| | Median Error | 9.02 | 7.24 | 13.87 | 8.67 | 4.69 | 11.59 |

Table 35: Unsupervised 3D pose estimation results for Corrupted-Pascal3D+ (frost) Part 1

| Method | Metric | plane | bicycle | boat | bottle | bus | car |
|---|---|---|---|---|---|---|---|
| (Wang et al., 2021a) | $\frac{\pi}{6}$ Accuracy | 57.6 | 65.9 | 39.0 | 77.0 | 86.8 | 95.8 |
| | $\frac{\pi}{18}$ Accuracy | 23.1 | 17.8 | 12.6 | 30.7 | 70.9 | 85.6 |
| | Median Error | 22.75 | 21.52 | 43.77 | 15.29 | 5.84 | 5.08 |
| **Ours** | $\frac{\pi}{6}$ Accuracy | 80.7 | 79.8 | 65.3 | 88.0 | 94.0 | 98.6 |
| | $\frac{\pi}{18}$ Accuracy | 51.9 | 27.8 | 38.7 | 49.9 | 89.7 | 94.4 |
| | Median Error | 9.62 | 16.07 | 15.22 | 10.02 | 3.02 | 3.26 |

Table 36: Unsupervised 3D pose estimation results for Corrupted-Pascal3D+ (frost) Part 2

| Method | Metric | chair | table | bike | sofa | train | tv |
|---|---|---|---|---|---|---|---|
| (Wang et al., 2021a) | $\frac{\pi}{6}$ Accuracy | 73.1 | 66.8 | 58.8 | 84.1 | 62.4 | 75.7 |
| | $\frac{\pi}{18}$ Accuracy | 31.6 | 38.8 | 12.5 | 35.6 | 47.5 | 24.6 |
| | Median Error | 15.99 | 14.94 | 24.66 | 12.51 | 11.38 | 18.29 |
| **Ours** | $\frac{\pi}{6}$ Accuracy | 87.9 | 80.1 | 81.5 | 89.7 | 90.5 | 82.9 |
| | $\frac{\pi}{18}$ Accuracy | 52.6 | 60.1 | 30.8 | 53.5 | 78.7 | 42.5 |
| | Median Error | 9.68 | 7.77 | 14.52 | 9.36 | 5.07 | 13.04 |

Table 37: Unsupervised 3D pose estimation results for Corrupted-Pascal3D+ (fog) Part 1

| Method | Metric | plane | bicycle | boat | bottle | bus | car |
|---|---|---|---|---|---|---|---|
| (Wang et al., 2021a) | $\frac{\pi}{6}$ Accuracy | 80.4 | 80.2 | 65.9 | 87.4 | 93.6 | 98.3 |
| | $\frac{\pi}{18}$ Accuracy | 46.4 | 27.4 | 36.3 | 47.0 | 89.5 | 94.2 |
| | Median Error | 10.82 | 16.62 | 15.8 | 10.52 | 3.24 | 3.63 |
| **Ours** | $\frac{\pi}{6}$ Accuracy | 85.5 | 81.6 | 70.6 | 89.3 | 95.9 | 99.2 |
| | $\frac{\pi}{18}$ Accuracy | 55.7 | 29.6 | 42.2 | 54.2 | 92.7 | 95.6 |
| | Median Error | 8.79 | 15.16 | 13.08 | 8.98 | 2.69 | 3.12 |

Table 38: Unsupervised 3D pose estimation results for Corrupted-Pascal3D+ (fog) Part 2

| Method | Metric | chair | table | bike | sofa | train | tv |
|---|---|---|---|---|---|---|---|
| (Wang et al., 2021a) | $\frac{\pi}{6}$ Accuracy | 86.2 | 80.1 | 73.6 | 92.6 | 87.6 | 78.7 |
| | $\frac{\pi}{18}$ Accuracy | 45.5 | 55.9 | 21.0 | 55.0 | 69.9 | 31.1 |
| | Median Error | 10.6 | 8.45 | 18.8 | 9.14 | 5.71 | 15.86 |
| **Ours** | $\frac{\pi}{6}$ Accuracy | 90.7 | 82.9 | 87.2 | 94.2 | 90.4 | 82.4 |
| | $\frac{\pi}{18}$ Accuracy | 58.5 | 65.0 | 34.7 | 61.1 | 79.0 | 44.1 |
| | Median Error | 8.73 | 6.97 | 13.26 | 8.16 | 4.74 | 11.72 |

Table 39: Unsupervised 3D pose estimation results for Corrupted-Pascal3D+ (contrast) Part 1

| Method | Metric | plane | bicycle | boat | bottle | bus | car |
|---|---|---|---|---|---|---|---|
| (Wang et al., 2021a) | $\frac{\pi}{6}$ Accuracy | 56.4 | 67.0 | 56.6 | 82.5 | 81.4 | 92.4 |
| | $\frac{\pi}{18}$ Accuracy | 18.2 | 14.9 | 25.6 | 42.3 | 61.1 | 80.3 |
| | Median Error | 24.48 | 22.18 | 23.49 | 11.96 | 7.22 | 5.55 |
| **Ours** | $\frac{\pi}{6}$ Accuracy | 87.7 | 81.9 | 70.4 | 89.2 | 94.0 | 99.2 |
| | $\frac{\pi}{18}$ Accuracy | 58.7 | 32.4 | 42.6 | 52.5 | 90.8 | 96.0 |
| | Median Error | 8.24 | 15.18 | 12.66 | 9.57 | 2.64 | 3.07 |

Table 40: Unsupervised 3D pose estimation results for Corrupted-Pascal3D+ (contrast) Part 2

| Method | Metric | chair | table | bike | sofa | train | tv |
|---|---|---|---|---|---|---|---|
| (Wang et al., 2021a) | $\frac{\pi}{6}$ Accuracy | 69.4 | 69.1 | 49.0 | 88.1 | 71.6 | 74.7 |
| | $\frac{\pi}{18}$ Accuracy | 26.1 | 38.2 | 7.4 | 45.7 | 45.5 | 28.5 |
| | Median Error | 15.8 | 14.94 | 30.86 | 10.62 | 11.29 | 17.95 |
| **Ours** | $\frac{\pi}{6}$ Accuracy | 90.5 | 83.0 | 86.4 | 95.2 | 91.0 | 81.8 |
| | $\frac{\pi}{18}$ Accuracy | 57.7 | 67.4 | 37.4 | 61.7 | 79.5 | 50.1 |
| | Median Error | 8.82 | 6.51 | 13.28 | 8.03 | 4.63 | 9.98 |

Table 41: Unsupervised 3D pose estimation for Corrupted-Pascal3D+ (elastic transform) Part 1

| Method | Metric | plane | bicycle | boat | bottle | bus | car |
|---|---|---|---|---|---|---|---|
| (Wang et al., 2021a) | $\frac{\pi}{6}$ Accuracy | 76.0 | 48.4 | 51.6 | 84.5 | 89.1 | 97.7 |
| | $\frac{\pi}{18}$ Accuracy | 39.9 | 10.2 | 19.8 | 36.9 | 81.8 | 92.7 |
| | Median Error | 13.68 | 30.82 | 27.66 | 12.87 | 4.64 | 4.02 |
| **Ours** | $\frac{\pi}{6}$ Accuracy | 84.9 | 81.6 | 71.6 | 88.8 | 94.5 | 98.9 |
| | $\frac{\pi}{18}$ Accuracy | 55.3 | 31.6 | 42.4 | 51.8 | 91.7 | 95.2 |
| | Median Error | 8.92 | 16.42 | 12.75 | 9.56 | 3.01 | 3.17 |

Table 42: Unsupervised 3D pose estimation for Corrupted-Pascal3D+ (elastic transform) Part 2

| Method | Metric | chair | table | bike | sofa | train | tv |
|---|---|---|---|---|---|---|---|
| (Wang et al., 2021a) | $\frac{\pi}{6}$ Accuracy | 63.2 | 72.8 | 42.9 | 92.0 | 87.6 | 74.1 |
| | $\frac{\pi}{18}$ Accuracy | 16.6 | 46.5 | 4.7 | 54.3 | 66.3 | 21.1 |
| | Median Error | 20.67 | 10.98 | 33.62 | 9.24 | 6.76 | 17.28 |
| **Ours** | $\frac{\pi}{6}$ Accuracy | 90.9 | 82.5 | 85.9 | 94.1 | 87.7 | 81.6 |
| | $\frac{\pi}{18}$ Accuracy | 55.1 | 63.2 | 33.3 | 61.7 | 69.7 | 44.3 |
| | Median Error | 9.32 | 7.13 | 14.3 | 8.05 | 6.3 | 11.65 |

Table 43: Unsupervised 3D pose estimation results for Corrupted-Pascal3D+ (pixelate) Part 1

| Method | Metric | plane | bicycle | boat | bottle | bus | car |
|--------|--------|-------|---------|------|--------|-----|-----|
| (Wang et al., 2021a) | $\frac{\pi}{6}$ Accuracy | 60.1 | 52.2 | 56.1 | 82.7 | 94.0 | 95.1 |
| | $\frac{\pi}{18}$ Accuracy | 25.9 | 9.5 | 28.4 | 39.1 | 88.0 | 90.1 |
| | Median Error | 20.61 | 28.77 | 23.4 | 12.37 | 4.12 | 4.03 |
| **Ours** | $\frac{\pi}{6}$ Accuracy | 86.0 | 82.6 | 73.2 | 88.4 | 94.9 | 98.9 |
| | $\frac{\pi}{18}$ Accuracy | 55.5 | 30.7 | 45.6 | 50.2 | 91.2 | 95.6 |
| | Median Error | 8.58 | 16.1 | 11.72 | 9.95 | 2.82 | 3.1 |

Table 44: Unsupervised 3D pose estimation results for Corrupted-Pascal3D+ (pixelate) Part 2

| Method | Metric | chair | table | bike | sofa | train | tv |
|--------|--------|-------|-------|------|------|-------|-----|
| (Wang et al., 2021a) | $\frac{\pi}{6}$ Accuracy | 82.8 | 78.9 | 43.4 | 93.9 | 75.5 | 81.2 |
| | $\frac{\pi}{18}$ Accuracy | 51.2 | 58.7 | 8.8 | 55.1 | 52.5 | 38.2 |
| | Median Error | 9.66 | 8.01 | 34.47 | 9.14 | 9.45 | 13.2 |
| **Ours** | $\frac{\pi}{6}$ Accuracy | 88.9 | 82.7 | 86.9 | 95.4 | 90.4 | 81.5 |
| | $\frac{\pi}{18}$ Accuracy | 56.9 | 65.2 | 34.2 | 59.8 | 79.2 | 44.0 |
| | Median Error | 9.03 | 6.9 | 13.23 | 8.43 | 5.07 | 11.38 |

Table 45: Unsupervised 3D pose estimation results for Corrupted-Pascal3D+ (speckle noise) Part 1

| Method | Metric | plane | bicycle | boat | bottle | bus | car |
|--------|--------|-------|---------|------|--------|-----|-----|
| (Wang et al., 2021a) | $\frac{\pi}{6}$ Accuracy | 44.8 | 59.1 | 37.3 | 47.9 | 81.6 | 95.7 |
| | $\frac{\pi}{18}$ Accuracy | 15.7 | 8.8 | 13.4 | 12.6 | 64.7 | 85.0 |
| | Median Error | 35.14 | 26.09 | 49.24 | 31.02 | 7.15 | 5.21 |
| **Ours** | $\frac{\pi}{6}$ Accuracy | 84.1 | 81.9 | 69.7 | 87.7 | 94.7 | 98.8 |
| | $\frac{\pi}{18}$ Accuracy | 53.5 | 27.9 | 42.2 | 51.7 | 91.2 | 95.1 |
| | Median Error | 9.22 | 15.93 | 13.24 | 9.79 | 2.93 | 3.22 |

Table 46: Unsupervised 3D pose estimation results for Corrupted-Pascal3D+ (speckle noise) Part 2

| Method | Metric | chair | table | bike | sofa | train | tv |
|---|---|---|---|---|---|---|---|
| (Wang et al., 2021a) | $\frac{\pi}{6}$ Accuracy | 63.8 | 56.5 | 66.2 | 71.8 | 70.5 | 75.7 |
| | $\frac{\pi}{18}$ Accuracy | 19.0 | 22.5 | 15.2 | 29.6 | 36.2 | 29.1 |
| | Median Error | 21.65 | 23.82 | 21.6 | 15.58 | 13.99 | 16.64 |
| **Ours** | $\frac{\pi}{6}$ Accuracy | 90.9 | 81.2 | 83.2 | 92.1 | 89.1 | 82.8 |
| | $\frac{\pi}{18}$ Accuracy | 52.6 | 62.9 | 38.0 | 55.0 | 76.7 | 46.7 |
| | Median Error | 9.47 | 7.28 | 12.66 | 8.84 | 4.83 | 10.66 |

Table 47: Unsupervised 3D pose estimation results for Corrupted-Pascal3D+ (gaussian blur) Part 1

| Method | Metric | plane | bicycle | boat | bottle | bus | car |
|---|---|---|---|---|---|---|---|
| (Wang et al., 2021a) | $\frac{\pi}{6}$ Accuracy | 42.2 | 55.5 | 39.7 | 81.3 | 62.0 | 86.0 |
| | $\frac{\pi}{18}$ Accuracy | 10.7 | 14.1 | 7.8 | 38.3 | 32.0 | 68.1 |
| | Median Error | 37.05 | 25.81 | 40.23 | 12.78 | 15.19 | 7.17 |
| **Ours** | $\frac{\pi}{6}$ Accuracy | 84.4 | 77.2 | 70.6 | 88.1 | 93.8 | 98.2 |
| | $\frac{\pi}{18}$ Accuracy | 51.7 | 28.4 | 41.1 | 47.8 | 90.0 | 94.2 |
| | Median Error | 9.62 | 16.62 | 13.38 | 10.51 | 3.08 | 3.31 |

Table 48: Unsupervised 3D pose estimation results for Corrupted-Pascal3D+ (gaussian blur) Part 2

| Method | Metric | chair | table | bike | sofa | train | tv |
|---|---|---|---|---|---|---|---|
| (Wang et al., 2021a) | $\frac{\pi}{6}$ Accuracy | 79.6 | 67.6 | 57.7 | 89.4 | 54.3 | 78.4 |
| | $\frac{\pi}{18}$ Accuracy | 39.7 | 38.1 | 11.6 | 42.8 | 28.6 | 32.5 |
| | Median Error | 12.45 | 14.38 | 25.99 | 11.47 | 23.2 | 15.19 |
| **Ours** | $\frac{\pi}{6}$ Accuracy | 88.9 | 83.7 | 83.0 | 95.0 | 89.6 | 82.3 |
| | $\frac{\pi}{18}$ Accuracy | 56.7 | 65.8 | 33.3 | 58.7 | 77.5 | 43.5 |
| | Median Error | 9.12 | 6.96 | 14.31 | 8.64 | 5.22 | 11.45 |

Table 49: Unsupervised 3D pose estimation results for Corrupted-Pascal3D+ (spatter) Part 1

| Method | Metric | plane | bicycle | boat | bottle | bus | car |
|---|---|---|---|---|---|---|---|
| (Wang et al., 2021a) | $\frac{\pi}{6}$ Accuracy | 58.3 | 64.0 | 41.6 | 64.5 | 88.3 | 96.0 |
| | $\frac{\pi}{18}$ Accuracy | 25.6 | 14.3 | 15.9 | 23.0 | 79.5 | 87.8 |
| | Median Error | 23.54 | 23.43 | 41.0 | 21.41 | 4.97 | 4.69 |
| **Ours** | $\frac{\pi}{6}$ Accuracy | 84.4 | 79.4 | 65.8 | 87.6 | 94.4 | 99.0 |
| | $\frac{\pi}{18}$ Accuracy | 52.9 | 28.4 | 38.3 | 50.5 | 91.0 | 95.2 |
| | Median Error | 9.44 | 16.77 | 15.38 | 9.96 | 3.03 | 3.25 |

Table 50: Unsupervised 3D pose estimation results for Corrupted-Pascal3D+ (spatter) Part 2

| Method | Metric | chair | table | bike | sofa | train | tv |
|---|---|---|---|---|---|---|---|
| (Wang et al., 2021a) | $\frac{\pi}{6}$ Accuracy | 73.1 | 56.6 | 63.5 | 78.0 | 82.1 | 70.4 |
| | $\frac{\pi}{18}$ Accuracy | 32.0 | 29.4 | 14.0 | 32.7 | 61.2 | 18.0 |
| | Median Error | 15.0 | 21.87 | 22.05 | 14.3 | 7.39 | 21.21 |
| **Ours** | $\frac{\pi}{6}$ Accuracy | 90.5 | 82.1 | 87.0 | 94.2 | 89.0 | 82.0 |
| | $\frac{\pi}{18}$ Accuracy | 56.5 | 62.8 | 34.0 | 59.5 | 76.9 | 44.9 |
| | Median Error | 8.81 | 7.45 | 13.68 | 8.24 | 5.28 | 11.27 |

