# OpenReview forum: "Source-Free and Image-Only Unsupervised Domain Adaptation for Category Level Object Pose Estimation"
_ICLR.cc/2024/Conference — ICLR 2024 poster_

### Official Review · Reviewer_wu5J · 2023-10-20

**Soundness:** 3 good
**Presentation:** 3 good
**Contribution:** 3 good
**Rating:** 6
**Confidence:** 3

**Summary:**

This paper introduces a new domain adaptation method for category-level object pose estimation. The method uses only RGB images, without relying on source domain data or 3D annotations in the target domain. The authors represent an object that belongs to a known category as a cuboid mesh and utilize an existing method to learn the vertex features. The features are iteratively updated in the target domain based on their proximity to corresponding image features. The experiments show much better performance in the target domain compared with the competitors without domain adaptation.

**Strengths:**

•	The authors handle the problem that neither depth information nor 3D annotations are available in the target domain, which is challenging yet important in real applications.

•	The intuition behind the presented domain adaptation method is that the local features, which represent specific parts of the object, are more robust than the global features in the target domain, which makes sense to me.

•	The presented method achieves impressive object pose estimation results in multiple datasets with different kinds of nuisance.

**Weaknesses:**

The majority of the figures in this paper such as Fig.2, Fig.2, and Fig.10, exhibit low quality. It would be better if the authors could consider revising them to enhance the clarity and resolution.

I am not familiar with the topic of domain adaptation, so I would not judge the novelty of this paper. Please refer to “Questions” for my concerns.

**Questions:**

•	To my understanding, in the experiments, the baseline models are evaluated in the target domain without domain adaptation. In this context, it is reasonable that those methods cannot generalize well in the target domain. I was wondering if there are some existing domain adaption approaches that can be applied to the baseline models. The evaluation would be more convincing, comparing NeMo + 3DUDA with a competitor such as NeMo + another domain adaption method.

•	The pre-render feature maps are generated from the source mesh. As the vertex features are not updated here, how to make sure those feature maps are reliable in the target domain? Are there situations in which the method might be stuck in local optima or even diverge? Conducting an ablation study on the pre-rendered feature maps would be valuable.

•	In Sec.3.2.1, the parameter $\delta$ seems crucial for effectively updating vertex features. The authors mentioned they chose a $\delta$ such that the majority of source domain features lie within the likelihood score. How to set $\delta$ in practice? Is it constant? Intuitively, as the vertex features are updated, one would expect the similarity between vertex features and image features to increase. Does it make more sense to update $\delta$ accordingly?

•	It seems the method is time-consuming due to the iterations and pre-rendering. However, the actual time consumption during testing is unclear.

---

> ### Author Response · Authors · 2023-11-18
> **Review Response**
>
> We thank the reviewer for their encouraging feedback.
>
> **W1. The majority of the figures in this paper such as Fig.2, Fig.2, and Fig.10, exhibit low quality. It would be better if the authors could consider revising them to enhance the clarity and resolution.**
>
> This point has also been raised by other reviewers. We have tried our best to address the presentation issues with our initial draft by improving the quality and clarity of the figures. We are happy to take any more suggestions to improve them even further. Please refer to the updated draft for updated and improved figures.
>
>
> **Q1. To my understanding, in the experiments, the baseline models are evaluated in the target domain without domain adaptation. In this context, it is reasonable that those methods cannot generalize well in the target domain. I was wondering if there are some existing domain adaption approaches that can be applied to the baseline models. The evaluation would be more convincing, comparing NeMo + 3DUDA with a competitor such as NeMo + another domain adaption method.**
>
> To the best of our knowledge, our method appears to be the only current source-free image-only UDA method for object pose estimation. It is unclear what kind of methodology we could combine with NeMo for UDA since that would be nontrivial and could be an interesting new research avenue. We do have baselines (P3D [a] and DMNT [b]), which are improvements over the NeMo model and claim to be robust to OOD as well. P3D uses synthetic and real data combinations and different training methods for robust 3D pose estimation. DMNT uses deformable meshes and neural textures for robust pose estimation.
>
> [a] Yang et. al. - Robust category-level 3d pose estimation from synthetic data
>
> [b] Wang et. al. - Neural textured deformable meshes for robust analysis-by-synthesis
>
>
> **Q2. The pre-render feature maps are generated from the source mesh. As the vertex features are not updated here, how to make sure those feature maps are reliable in the target domain? Are there situations in which the method might be stuck in local optima or even diverge? Conducting an ablation study on the pre-rendered feature maps would be valuable.**
>
> We discuss this exact problem in Section 3.2 (Multi-Pose Initialization) and provide an ablative example in Figure 7. The source model does sometimes get stuck in local optima when evaluated on the target domain data, and therefore the estimated result may depend on the pose initialization. Therefore, we do multi-pose initializations for our method, and use the final estimates to update only robust, local vertex features irrespective of the global object pose estimated. We have added an ablation for multi-pose initializations in our updated draft (Table 4).
>
> **Q3. In Sec.3.2.1, the $\delta$ parameter
> seems crucial for effectively updating vertex features. The authors mentioned they chose a $\delta$ such that the majority of source domain features lie within the likelihood score. How to set in practice? Is it constant? Intuitively, as the vertex features are updated, one would expect the similarity between vertex features and image features to increase. Does it make more sense to update $\delta$ accordingly?**
>
> For our experiments, we empirically find that a normalized value between 0.85-0.9 for the parameter works sufficiently well for our model and approximates the coverage of the majority of source domain vertex features. Tightening the parameter during model updates helps the model learn faster. We thank the reviewer for their suggestion! Here is the ablative experiment for bus category on OOD-CV dataset with different threshold values:
>
> | Threshold value  |  Epochs range for convergence |
> |------------------|-------------------------------|
> |        0.8       |            120-150            |
> |       0.85       |             85-100            |
> | Dynamic(.85-.99) |             50-70             |
>
> **Q4. It seems the method is time-consuming due to the iterations and pre-rendering. However, the actual time consumption during testing is unclear.**
>
> Thanks for pointing out that this should have been discussed in the paper. Evaluation time for our model (if not doing batched inference) is ~0.33 seconds/sample on a Nvidia RTX 2060 GPU. We have added it in the paper.

---

> > ### Comment · Reviewer_wu5J · 2023-11-22
> >
> > Thanks for the response. I have no other questions but it's better to further improve the presentation.

---

> > > ### Author Response · Authors · 2023-11-22
> > >
> > > Dear Reviewer,
> > >
> > > We have gone to a great length to update the draft and address your comments. It would be tremendous feedback to us if you can share what else you do not like about the presentation, we will fix those issues immediately. And if you feel that all technical questions are answered, it would be encouraging if you could kindly increase the score. Thank you!

---

### Official Review · Reviewer_PE77 · 2023-10-30

**Soundness:** 3 good
**Presentation:** 3 good
**Contribution:** 3 good
**Rating:** 6
**Confidence:** 4

**Summary:**

This paper addresses the task of category-level 3D pose estimation within the setting of source-free and image-only unsupervised domain adaptation. The authors introduce a novel method called 3DUDA, which is developed based on the observation of the invariance of object local parts and is supported by theoretical insights. 3DUDA utilizes a learnable cuboid feature matrix to represent an object category, and assesses the accuracy of a pose by comparing the feature map of the test image with the one rendered from the categorical cuboid feature matrix using the pose. This render and compare approach enables domain adaptation by iteratively updating the categorical feature matrix and fine-tuning the source model. Experimental results demonstrate the effectiveness of the proposed method.

**Strengths:**

- The proposed method, 3DUDA, is developed based on the common observation that certain object parts exhibit invariance across out-of-distribution (OOD) scenarios, and utilizes the categorical learnable cuboid meshes to effectively capture and store the part features at each vertex.
- To achieve domain adaptation, 3DUDA employs an iterative process that involves updating the features of categorical meshes and fine-tuning the source model through feature-level render and compare optimization.
- The paper is well-written and presents its ideas in a clear and understandable manner. It is accompanied by comprehensive supplementary materials and theoretical results, which greatly enhance the persuasiveness of the paper.

**Weaknesses:**

- It is recommended to evaluate the proposed method on commonly used datasets for category-level pose estimation, such as REAL275 [1] or Wild6D [2].

- It would be beneficial to include relevant works [3,4,5,6] in the paper to provide a comprehensive overview of the existing literature in the field.

[1] Wang et al., Normalized object coordinate space for category-level 6d object pose and size estimation. CVPR2019.

[2] Fu et al., Category-Level 6D Object Pose Estimation in the Wild: A Semi-Supervised Learning Approach and A New Dataset.

[3] Lin et al., Category-level 6D object pose and size estimation using self-supervised deep prior deformation networks. ECCV2022.

[4] He et al., Towards Self-Supervised Category-Level Object Pose and Size Estimation.

[5] Zhang et al., Self-Supervised Geometric Correspondence for Category-Level 6D Object Pose Estimation in the Wild. ICLR2023.

[6] Goodwin et al., Zero-Shot Category-Level Object Pose Estimation. ECCV2022.

**Questions:**

- How does 3DUDA address the issue of minimizing the impact of translation and object size on neural feature rendering?
- What values are set for the number $R$ of vertices of the mesh and the hyperparameter of $N$ of the clutter model? How do they  impact the performance of the method?
- The format of citations in Table 1 does not align with the citation format used in the rest of the paper.

---

> ### Author Response · Authors · 2023-11-18
> **Review Rebuttal (Part 1)**
>
> We thank the reviewer for their time.
>
> **W1. It is recommended to evaluate the proposed method on commonly used datasets for category-level pose estimation, such as REAL275 or Wild6D**.
>
> - Our focus in this work is largely on outdoor setups (e.g. OOD-CV) due to relatively more diverse OOD conditions (weather, lighting, context, etc.) available in such datasets.
>
> - Both WILD6D+ and REAL275 datasets are indoor RGBD datasets in NOCS [1] format, which are used primarily for robotic applications and have 5-6 categories. Historically, works have either focused on datasets like Pascal3D+, ObjectNet3D+ or REAL275, WILD6D+ and we follow the same convention. This is likely due to incompatibility between the camera parameters and data formats of the two kinds of datasets. It is not trivial to convert one to another (or convert models of one kind to another) and is beyond the scope of this review period due to technical and time-related limitations. We believe that combining these two baseline branches would strengthen the research field, but it remains a future work.
>
> - Most models [3,4,5] using these datasets rely on the depth channel to run PnP or combine depth with NOCS to solve pose so they cannot be trivially modified as an RGB model (to compare with our method) and require extensive work.
>
> - We therefore follow the convention of datasets used in previous works related to our pose estimation method.
>
> As an alternative for the Synthetic to Real evaluation done in REAL275 and WILD6D+ benchmarks, here are the results for synthetic to real dataset using the SyntheticP3D source domain dataset[2], which consists of 6 synthetic objects. The real dataset is Pascal3D+.
>
> | Methods           | $\pi/6$ Accuracy | $\pi/18$ Accuracy |
> |-------------------|:----------------:|:-----------------:|
> | Resnet-50 General |       53.5       |        13.2       |
> | P3D               |       76.5       |        41.3       |
> | NeMo              |       71.8       |        39.5       |
> | Ours              |       88.9       |        66.7       |
>
>
> [1] Wang et al. - Normalized Object Coordinate Space for Category-Level 6D Object Pose and Size Estimation.
>
> [2] Yang et al. - Robust Category-Level 3D Pose Estimation from Synthetic Data.
>
> [3] Lee et al. - TTA-COPE: Test-Time Adaptation for Category-Level Object Pose Estimation.
>
> [4] Zhang et al. - Self-Supervised Geometric Correspondence for Category-Level 6D Object Pose Estimation in the Wild.
>
> [5] Liu et al. - IST-Net: Prior-free Category-level Pose Estimation with Implicit Space Transformation.
>
> **W2. It would be beneficial to include relevant works [3,4,5,6] in the paper to provide a comprehensive overview of the existing literature in the field.**
>
> Thanks for pointing out the works. We have added the mentioned references in the updated draft.

---

> > ### Author Response · Authors · 2023-11-18
> > **Review Response (Part 2)**
> >
> > **Q1. How does 3DUDA address the issue of minimizing the impact of translation and object size on neural feature rendering?**
> >
> > The problem that we are focusing on in this work is UDA for 3D pose estimation, therefore, we are not evaluating our models for translation. Many works utilize object detectors for localizing objects and estimating translation/scale, and our method can be extended in a similar fashion. We do not evaluate our model for scale robustness but we can optimize the distance from camera parameters and therefore are able to deal with the object scale changes in our current datasets.
> >
> > **Q2. What values are set for the number of vertices of the mesh and the hyperparameter of $N$ of the clutter model? How do they impact the performance of the method?**
> >
> > The number of vertices used is approximately 1000 for all categories. We take 1000 as the number of vertices, since the distance between nearest vertices after projection is roughly the same as the stride of the backbone network. Halving the number of vertices does not have a significant effect on the results. However, doubling the number of vertices leads to significant deterioration. This is likely due to a large number of vertices corresponding to the same feature vector, which complicates the feature learning process. Here is an ablation analysis -
> >
> > | Number of vertices | Median Error | $\pi/18$ Accuracy | $\pi/6$ Accuracy |
> > |:------------------:|:------------:|:-----------------:|:----------------:|
> > |         500        |     3.56     |        90.1       |       93.9       |
> > |        1000        |     3.03     |        91.0       |       94.4       |
> > |        2000        |     20.63    |        50.9       |       66.3       |
> >
> > Our method is not sensitive to the change in the number of clutter models. This could be attributed to the fact that our similarity threshold can be construed as a clutter model with uniform distribution. We do see small improvements as the number of clutter models is drastically increased, but that comes at a high computational cost. Here is an ablation analysis for the same (Table 5 in the updated draft).
> >
> > | Clutter ($N$) | Median Error | $\pi/18$ Accuracy | $pi/6$ Accuracy |
> > |:-------------:|:------------:|:-----------------:|:---------------:|
> > |       1       |     3.01     |        90.1       |       95.1      |
> > |       5       |     3.03     |        91.1       |       94.4      |
> > |       20      |     2.99     |        92.3       |       95.5      |
> > |       35      |     2.79     |        91.7       |       94.5      |
> >
> > **Q3. The format of citations in Table 1 does not align with the citation format used in the rest of the paper.**
> >
> > We apologize for this inconsistency, which has now been fixed.

---

> > > ### Comment · Reviewer_PE77 · 2023-11-22
> > > **Post-rebuttal assessment**
> > >
> > > I appreciate the authors' response, which addresses my concerns. Consequently, I will maintain my positive rating.

---

### Official Review · Reviewer_xesL · 2023-11-01

**Soundness:** 3 good
**Presentation:** 1 poor
**Contribution:** 2 fair
**Rating:** 6
**Confidence:** 4

**Summary:**

This paper addresses the challenge of unsupervised category-level pose estimation in a target domain using only RGB images, without access to source domain data or 3D annotations.
The authors introduce a method that adapts to a target domain, even when it is complicated by nuisances, without requiring 3D or depth data.
They represent object categories with simple cuboid meshes and use a generative model of neural feature activations at each mesh vertex.
They focus on updating local mesh vertex features based on their proximity to corresponding features in the target domain, even when the global pose is incorrect.
The key insight is the stability of specific object sub-parts across different scenarios, which allows for effective model updates.

**Strengths:**

1. This paper suggests effective render-and-compare adaptation pipeline for unsupervised domain adaptation for category-level object pose estimation task.

2. Its proposed method shows the state-of-the-art performance in various corruption scenarios compared to some of the previous methods.

3. The methodology part is intuitive and easy to follow text-wise.

**Weaknesses:**

1. Poor presentation.
I think this paper holds good insights and corresponding technological contributions.
Yet, the presentation of the whole paper is relatively poor, making it hard to follow the overall message.
Figure placement is not well-aligned with the text context.
Figure 3 seems to be a very important description of explaining one of the main contributions of this paper to match sub-vertices, while there is no mention in the main manuscript referring this.
Moreover, the main manuscript refers to figures in appendix very often, which is very inconvenient to read, while some of these figures seem important enough to be contained in the main paper for effective description. (ex, Figure 5)
I believe that ablation studies regarding several design choices of the proposed method should be contained in the main manuscript as well, since they can effectively validate the authors claim.

**Questions:**

1. Baseline methods and other datasets
Except OOD-CV results, only previous baseline is NeMo. Can the authors explain why there can't be other methods in this comparison? Being better than only one baseline is not enough to claim the superiority of the proposed method.
Also, while I acknowledge that the paper mainly focuses on data corruption scenarios, is it possible to compare this type of approach in conventional category-level object pose benchmarks like CAMERA and REAL275 datasets provided by NOCS? I believe it would strengthen the authors' motivation if it can be generally applied to conventional syn-to-real UDA scenarios.

2. GT reliability
Ground truth pose illustrated in Figure 1 and 5 seems to be not perfectly aligned with the image. Can the authors explain how these GTs are obtained, and how they are utilized? Are they only used for evaluation?

**Details Of Ethics Concerns:**

.

---

> ### Author Response · Authors · 2023-11-18
> **Review Response (Part 1)**
>
> We thank the reviewer for their valuable feedback. In response to their comments:
>
> **W1a. Poor presentation.  I think this paper holds good insights and corresponding technological contributions. Yet, the presentation of the whole paper is relatively poor, making it hard to follow the overall message. Figure placement is not well-aligned with the text context. Figure 3 seems to be a very important description of explaining one of the main contributions of this paper to match sub-vertices, while there is no mention in the main manuscript referring this.**
>
> We regret that the presentation of our initial draft was deemed poor by the reviewer. According to their suggestions, we have updated our paper. We have reframed the main content to make it easier to follow, improved the quality and placement of figures and made sure that all figures are properly referenced in the text. We request the reviewer to take a look at the newer version to see if the presentation issues are addressed.
>
> **W1b: Moreover, the main manuscript refers to figures in appendix very often, which is very inconvenient to read, while some of these figures seem important enough to be contained in the main paper for effective description. (ex, Figure 5) I believe that ablation studies regarding several design choices of the proposed method should be contained in the main manuscript as well, since they can effectively validate the authors claim.**
>
> With our restructuring of the paper, we have moved some important details up from the appendix to the main text as we agree with the reviewer that some of them are too important to be deferred to the appendix. We have updated old Figure 3 and moved old Figure 5 to the main draft (refer to Figures 1 and 3 in the updated draft). Unfortunately, numerous experimental analysis performed in this work can not all fit in the space constraints of the main text. We hope that the current restructuring in the new draft bolsters the contributions that we made with this work. We are happy to take more suggestions regarding the same and address any more issues that the reviewer thinks still exist.
>
> **Q1a. Baseline methods and other datasets Except OOD-CV results, only previous baseline is NeMo. Can the authors explain why there can't be other methods in this comparison? Being better than only one baseline is not enough to claim the superiority of the proposed method.**
>
> We thank the reviewer for drawing attention to this point. We have added results for other baselines' extended results tables in our updated draft (Table 6 and Table 7 (page 22 and 23)) which includes four more comparative methods for every experiment. As can be gleaned from these baselines, like OOD-CV, our method performs exceedingly well even when dealing with extreme UDA setups.

---

> ### Author Response · Authors · 2023-11-18
> **Review Response (Part 2)**
>
> **Q1b. Also, while I acknowledge that the paper mainly focuses on data corruption scenarios, is it possible to compare this type of approach in conventional category-level object pose benchmarks like CAMERA and REAL275 datasets provided by NOCS? I believe it would strengthen the authors' motivation if it can be generally applied to conventional syn-to-real UDA scenarios.**
>
> Thank you for drawing our attention to CAMERA and REAL275 datasets.
>
>  - Our focus in this work is largely on real, outdoor setups (e.g. OOD-CV) due to relatively more diverse OOD conditions (weather, lighting, context, etc.) available in such datasets.
>
> - Conventionally, 3D pose estimation works have either focused on setups and datasets like PASCAL3D+, ObjectNet3D or on more robotic application based synthetic to real datasets like REAL275, CAMERA and WILD6D+. This is due to some distinct technical and usage difference between these dataset setups. It’s non-trivial to convert the camera calibrations, annotations and dataformats from one to another and would require a significant amount of time and is likely out of scope for the review period. To the best of our knowledge, there is no recent work which has used both of these kinds of datasets together, and we believe this would be a good future work to combine these two types of datasets into one format.
>
> - Both CAMERA and REAL275 datasets are indoor RGBD datasets, which are used primarily for robotic applications and have 5-6 categories. Most models[3,4,5] using these datasets[1] rely on the depth channel to run PnP or combine depth with NOCS[1] to solve pose so they cannot be trivially modified as an RGB model (to be compared with our method) and require extensive work.
>
> - As recommended by the reviewer - for a conventional synthetic to real analysis, here are results for image only unsupervised domain adaptation from SyntheticP3D source domain dataset[2] (which has 6 synthetic object classes) to real Pascal3D+ target domain dataset.
>
>
> | Methods           | $\pi/6$ Accuracy | $\pi/18$ Accuracy |
> |-------------------|:----------------:|:-----------------:|
> | Resnet-50 General |       53.5       |        13.2       |
> | P3D               |       76.5       |        41.3       |
> | NeMo              |       71.8       |        39.5       |
> | Ours              |       88.9       |        66.7       |
>
> [1] Wang et al. - Normalized Object Coordinate Space for Category-Level 6D Object Pose and Size Estimation.
>
> [2] Yang et al. - Robust Category-Level 3D Pose Estimation from Synthetic Data.
>
> [3] Lee et al. - TTA-COPE: Test-Time Adaptation for Category-Level Object Pose Estimation.
>
> [4] Zhang et al. - Self-Supervised Geometric Correspondence for Category-Level 6D Object Pose Estimation in the Wild.
>
> [5] Liu et al. - IST-Net: Prior-free Category-level Pose Estimation with Implicit Space Transformation.
>
> **Q2. GT reliability Ground truth pose illustrated in Figure 1 and 5 seems to be not perfectly aligned with the image. Can the authors explain how these GTs are obtained, and how they are utilized? Are they only used for evaluation?**
>
> We want to clarify that the CAD objects in old Figure 1 and 5 (updated Figure 3) are for visualisation only. We use Shapenet3D models and rotate them according to the dataset’s annotations (GT) and model’s (Ours, NeMo) pose estimation results. There might be some noise in GT pose annotation as well the visualization setup which might be the reason for non-perfect alignment. The images as well as their ground truth annotations (GTs) in the figures are obtained from the OOD-CV dataset.

---

> ### Author Response · Authors · 2023-11-22
> **Discussion Period End**
>
> Dear Reviewer,
>
> We have updated our paper's presentation according to your comments and have covered all the technical questions that you've asked. It'll be of vastly beneficial to us if you can tell us what additional improvements you'd like to see in our paper, so we can make them before the discussion period ends. If you are satisfied with our method and believe we have covered your technical questions, it'll be reassuring if you can kindly update our paper's rating to reflect the same.

---

> > ### Comment · Reviewer_xesL · 2023-11-23
> >
> > I thank the authors for their response. It resolves my concerns and questions. I change my rating to 6.

---

### Official Review · Reviewer_KkCS · 2023-11-08

**Soundness:** 2 fair
**Presentation:** 2 fair
**Contribution:** 2 fair
**Rating:** 6
**Confidence:** 3

**Summary:**

This paper study the problem of source-free unsupervised category-level pose estimation from only RGB images to a target domain without any access to source domain data or 3D annotations during adaptation. The author propose a new pipline which focus on the  individual local mesh vertex features and utilize their pose ambiguity to iteratively update them based on their prox- imity to corresponding features in the target domain even when the global pose is not correct. The proposed method shows good results.

**Strengths:**

1. The proposed method outperform previous approaches by a large margin
2. The author evaluate their model on real world nuisances like shape, texture, occlusion, etc. as well as image corruptions and show the robustness of proposed method

**Weaknesses:**

1. As mentioned in the article, an observation is that global information is noisy, but some local details are robust. I hope there is rigorous explanation and quantitative analysis here to support this hypothesis.
2. Ablation is not enough, (e.g., Ablation on top-n
3. I think this method is similar to the iterative optimization often used in instance-level post-processing. It is unfair to compare this method with other single forward methods.
4. I'm not sure if it's right to say "normalized real-valued feature activations" ? (
The fourth line from the bottom of the third page)
5. The figures in this article are very rough and difficult to understand.

**Questions:**

Please refer to the weaknesses.

---

> ### Author Response · Authors · 2023-11-18
> **Review Response and Clarifications**
>
> We thank the reviewer for their valuable feedback. In response to their comments:
>
> **W1. As mentioned in the article, an observation is that global information is noisy, but some local details are robust. I hope there is rigorous explanation and quantitative analysis here to support this hypothesis.**
>
> Section 3.3 provides theoretical justification of how local robustness helps the domain adaptation problem. To see whether this local robustness is observed in real datasets, we calculated the percentage of correctly detected vertices in the target domain. In Figure 1 (in the updated draft), we see that this value has a minimum of 5-20% robust vertex features in the object category “Car”. After adaptation using our method, this percentage increases significantly. This is further illustrated for other object categories in the appendix (Figures 5, 9, 10). If some specific quantitative analysis is missing, we request the reviewer to point it out so that we can update it.
>
> **W2. Ablation is not enough, (e.g., Ablation on top-n**
>
> It is unclear what the reviewer means by missing ablation studies **(top-n)** in our 3-D pose estimation problem. We believe that we have ablated all major design choices and method motivations in Section A.5. In summary, our ablation (Section A5) includes ablation of
>
> a. Our motivation and observations regarding local part robustness (Figure 5, 9, 10),
>
> b. The efficacy of the Selective Vertex Feature Update,
>
> c. False Positives in vertex-Feature Similarity,
>
> d. Effect of the learnable concentration parameter,
>
> e. Multi-Pose Initialization,
>
> f. Quality of vertex Features after Selective Adaptation.
>
> In addition, we have added the following more ablations in the revised version:
>
> g. Effect of number of clutter features hyperparameter,
>
> h. Additional ablation figures to prove the local part robustness observation mentioned in the previous response paragraph. (Figures 9 and 10)
>
> If the reviewer is referring to the top-n multipose initializations, we do observe faster convergence with increasing number of initial poses. However, during multipose initialization, we only consider estimated poses whose rotational matrix distance (refer to Section 4 (Metrics paragraph)) from one another is at least $\pi/5$. If the estimated poses are less than the threshold for all, we only consider the pose with the highest similarity to the image feature vector. We did not see any significant improvement in accuracy using more than 5 initial poses for our experiments. Here are ablation results for the car category for different numbers of pose initializations when adapted on OOD-CV (Combined dataset).
>
> | Number of initialized poses | Epochs range for convergence | $\pi/18$ Accuracy |
> |-----------------------------|------------------------------|-------------------|
> |              1              |            130-170           |        95.3       |
> |              3              |            100-120           |        97.1       |
> |              5              |             50-90            |        97.1       |
>
> If there are specific ablation studies that the reviewer feels are missing, we would be happy to run them.
>
> **W3. I think this method is similar to the iterative optimization often used in instance-level post-processing. It is unfair to compare this method with other single forward methods.**
>
> It will be helpful if the reviewer can clarify which methods or provide citations of the works that use iterative optimization and are similar to our UDA method. To the best of our knowledge, our method is the first method to do source-free, image-only unsupervised domain adaptation for 3D object pose estimation. If the reviewer refers to the render-and-compare optimization used in our model, we would like to clarify that this is indeed a common optimization method used for a number of pose estimation methods such as [a,b,c] (this is not our contribution) and it is standard practice to compare such methods with feedforward methods as can be seen in these works.
>
> [a] Li et al. - DeepIM: Deep Iterative Matching for 6D Pose Estimation
>
> [b] Wang et al. - NeMo: Neural Mesh Models of Contrastive Features for Robust 3D Pose Estimation
>
> [c] Chen et al. - Category level object pose estimation via neural analysis-by-synthesis
>
> **W4. I'm not sure if it's right to say "normalized real-valued feature activations" ? ( The fourth line from the bottom of the third page)**
>
> The phrase "normalized real-valued feature activations"  refers to feature activations of a neural model which are real numbers and are normalized to have unit norm. We request the reviewer to be more specific as to why they believe the usage is incorrect.
>
> **W5. The figures in this article are very rough and difficult to understand.**
>
> We have updated the visual clarity of our figures in the main draft (refer to Figures 1, 2 and 3) and modified them to be easier to understand. We hope that the revised figures address this concern of the reviewer.

---

> > ### Comment · Reviewer_KkCS · 2023-11-22
> >
> > Thanks for authors detailed response, my concerns are addressed. I will lift my rating to 6

---

> ### Author Response · Authors · 2023-11-22
> **Discussion Deadline**
>
> Dear Reviewer,
>
> As the reviewer-author period comes to an end, we are yet to receive any response or acknowledgment from you regarding our response to your initial review. We have also asked some clarifications regarding your initial review so we can frame our response better. Please let us know if you have any further questions or if our clarifications and paper updates are satisfactory for an improved rating.

---

### Author Response · Authors · 2023-11-18
**General Response to Reviews**

We thank all the reviewers for their comments and observations.

In this work, we address “a challenging yet important real application“ (reviewer **wu5J**) concerning source-free and image-only 3D object pose estimation. Our method, 3DUDA, “achieves impressive object pose estimation results in multiple datasets with different kinds of nuisance” (reviewer **wu5J**) with “state-of-the-art performance in various corruption scenarios” (reviewer **xesL**). We also would like to thank the reviewers for pointing out that the “methodology is intuitive and easy” (reviewer **xesL**) to understand.

The key insight of our method lies in the observation that certain “object parts exhibit invariance across out-of-distribution (OOD) scenarios” (reviewers **PE77**, **xesL**) which is explained as Local Pose Ambiguity and Local Part Robustness in our work (see *Figure 1* in the updated draft) that allows one to update our source neural mesh model even when the global pose is incorrectly evaluated.

Our work is supported by "comprehensive supplementary material and theoretical insights" (reviewer **PE77**). We also evaluate our model on a novel extreme UDA setup, where we combine different kinds of OOD nuisance, including occlusion in the target domain.

In our updated draft, we have made an effort to further simplify our method. We have updated our draft's figures and improved the presentation of the methodology and the placement of results and figures. We have also shifted some important figures and results from the Appendix to the main text (Refer to *Figures 1 and 3*) to aid the readers in better understanding of our work. In addition to our extensive results and ablation, we have added more experimental analysis and ablations (refer to *Tables 4, 5, 6, 7* and *Figures 9, 10*).

We also clarify our choice of datasets in this work to address comments from reviewers **xesL** and **PE77**. We follow the baseline conventions of all major previous works connected to our method. Datasets like CAMERA, REAL275, WILD6D+, etc. and methods that use them cannot be trivially converted to pose estimation works utilizing Pascal3D+, ObjectNet3D, etc. datasets due to significant technical differences. To the best of our knowledge, there is no recent pose estimation work that uses both types of datasets. Although it would be beneficial for the community if we could combine these datasets into one common format, their usage in our work is out of scope due to technical and time-related constraints. Additionally, most methods that use CAMERA, etc. baselines often require depth data or point cloud, etc. and there is no trivial way to modify them for RGB only setup in order to compare with our work. However, since these baselines primarily concern Synthetic-to-Real transfer, we show results for image-only unsupervised domain adaptation from the SyntheticP3D dataset [1] (which has six synthetic object classes) to the real Pascal3D+ target domain dataset, as follows.

| Methods           | $\pi/6$ Accuracy | $\pi/18$ Accuracy |
|-------------------|:----------------:|:-----------------:|
| Resnet-50 General |       53.5       |        13.2       |
| P3D               |       76.5       |        41.3       |
| NeMo              |       71.8       |        39.5       |
| Ours              |       88.9       |        66.7       |




[1] Yang et al. - Robust Category-Level 3D Pose Estimation from Synthetic Data.

---

> ### Author Response · Authors · 2023-11-22
> **Follow Up to General Response**
>
> We are happy that reviewers - who have engaged in a conversation with us - have acknowledged that we have addressed all their concerns and questions. We find their positive outlook towards our work sincerely encouraging.
>
> For the reviewer (xesL) who hasn't been able to engage with us in a conversation, we've ensured that all your concerns have also been addressed through our comprehensive response.

---

### Meta-Review · Area_Chair_T8DB · 2023-12-09

**Metareview:**

The paper introduces a novel method for source-free unsupervised 3D pose estimation from RGB images, using cuboid meshes and a generative model to adapt to target domains without 3D data or annotations, focusing on invariant object subparts for model updates and significantly improving pose estimation accuracy. All reviewers agree to accept the paper after the rebuttal. The author did a good job in rebuttal. After looking at the comments and paper, the AC agrees to accept the paper.

**Justification For Why Not Higher Score:**

While the reviewers agree to accept, there are still reservations on presentation and results.

**Justification For Why Not Lower Score:**

All reviewers agree to accept.

---

### Decision · Program_Chairs · 2024-01-16

Accept (poster)